# Workflow for high-dimensional flow cytometry analysis of T cells from tumor metastases

Cristina Faccani[1,2], Gianluca Rotta[3] , Francesca Clemente[4] , Maya Fedeli[1,5], Danilo Abbati[6], Francesco Manfredi[6], Alessia Potenza[6], Achille Anselmo[7], Federica Pedica[8], Guido Fiorentini[9], Chiara Villa[7], Maria P Protti[4], Claudio Doglioni[8,5], Luca Aldrighetti[9], Chiara Bonini[6,5], Giulia Casorati[1] , Paolo Dellabona[1], Claudia de Lalla[1]

We describe a multi-step high-dimensional (HD) flow cytometry workflow for the deep phenotypic characterization of T cells infiltrating metastatic tumor lesions in the liver, particularly derived from colorectal cancer (CRC-LM). First, we applied a novel flow cytometer setting approach based on single positive cells rather than fluorescent beads, resulting in optimal sensitivity when compared with previously published protocols. Second, we set up a 26-color based antibody panel designed to assess the functional state of both conventional T-cell subsets and unconventional invariant natural killer T, mucosal associated invariant T, and gamma delta T ($\gamma\delta$T)-cell populations, which are abundant in the liver. Third, the dissociation of the CRC-LM samples was accurately tuned to preserve both the viability and antigenic integrity of the stained cells. This combined procedure permitted the optimal capturing of the phenotypic complexity of T cells infiltrating CRC-LM. Hence, this study provides a robust tool for high-dimensional flow cytometry analysis of complex T-cell populations, which could be adapted to characterize other relevant pathological tissues.

## Introduction

High dimensional (HD) flow cytometry enables the analysis of a broad range of surface and intracellular antigens, by interrogating high numbers of cells and characterizing rare populations for the expression of multiple parameters (Palit et al, 2019). These technological advances have enhanced the deep phenotype dissection and complex network characterization in heterogeneous tissues at a single cell level. The component characterization of the tumor microenvironment (TME), aimed at evaluating the heterogeneous cellular network that conditions anti-tumor immune responses

(Hiam-Galvez et al, 2021; Vitale et al, 2021), is of particular interest. Cancer, myelo-monocytic (Kwak et al, 2020; Tcyganov et al, 2021), and regulatory T (Treg) cells (De Simone et al, 2016; Ohue & Nishikawa, 2019) are widely recognized suppressors of anti-tumor T-cell immune response. Effector T cells infiltrating the TME progressively differentiate toward an "exhausted" state, expressing high levels of inhibitory molecules that impair their ability to control tumor progression (Wherry & Kurachi, 2015; Blank et al, 2019). The crosstalk between CD4+ follicular helper T cells (T$_{FH}$) and B cells in the TME can lead to the organization of tertiary lymphoid structures, which can have opposite prognostic values depending on the tumor type (Protti et al, 2014; Sautès-Fridman et al, 2020; Noël et al, 2021). Furthermore, in addition to MHC-restricted conventional T cells, non-MHC–restricted T cells, defined as unconventional ones, are also implicated in spontaneous tumor immune-surveillance (Godfrey et al, 2010; Mori et al, 2016; Gorini et al, 2017; Lawand et al, 2017; Cortesi et al, 2018; Li et al, 2020). The major known unconventional populations are invariant natural killer T (iNKT), mucosal associated invariant T (MAIT), and gamma delta T ($\gamma\delta$T) cells, which are specific for non-peptide antigens presented or associated with the non-polymorphic CD1d, MR1, or butyrophilin 3A1 (BTN3A1) molecules, respectively, and together can account for up to 30% of total T lymphocytes particularly in the liver (Godfrey et al, 2015). This complex composition of immune populations in the TME may also be differentially organized from primary to metastatic sites in an organ-dependent manner, impacting the therapeutic outcome (Peinado et al, 2017). Collectively, this premise depicts a very complex cellular contexture in the TME, whose understanding can greatly help both in deciphering the mechanisms of tumor progression and in the design of new therapeutic strategies.

CRC is the third cause of cancer death worldwide, particularly in its metastatic forms that mostly target the liver (Xi and Xu, 2021). We are interested in the deep profiling of the T-cell immune landscape of hepatic metastases of CRC by HD flow cytometry. In the present

[1]Experimental Immunology Unit, Ospedale San Raffaele Scientific Institute, Milan, Italy   [2]School of Medicine and Surgery, University of Milano-Bicocca, Monza, Italy   [3]BD Biosciences Europe, Milan, Italy   [4]Tumor Immunology Unit, Ospedale San Raffaele Scientific Institute, Milan, Italy   [5]Università Vita-Salute San Raffaele, Milan, Italy   [6]Experimental Hematology Unit, Ospedale San Raffaele Scientific Institute, Milan, Italy   [7]Flow Cytometry Resource, Advanced Cytometry Technical Applications Laboratory (FRACTAL) Ospedale San Raffaele Scientific Institute, Milan, Italy   [8]Department of Experimental Oncology, Pathology Unit, Ospedale San Raffaele Scientific Institute, Milan, Italy   [9]Hepatobiliary Surgery, Ospedale San Raffaele Scientific Institute, Milan, Italy

Correspondence: delalla.claudia@hsr.it

work, we describe a novel methodological workflow for HD flow cytometric analysis of tumor-infiltrating T cells, including conventional effector, Treg, T$_{FH}$, and unconventional T-cell subsets from CRC-derived liver metastases (CRC-LM), encompassing: (1) application of a recently developed cell-based calibration protocol for optimizing the flow cytometer setting; (2) design of a novel 26-color based T-cell panel sampling both conventional and unconventional T-cell subsets; (3) implementation of a protocol to isolate single cell suspensions from surgically resected CRC and CRC-LM. The combination of these accurate experimental procedures with the data analysis by cytoChain (Manfredi et al, 2021), a recently described in silico workflow for flow high-dimensional analysis, offers a reliable tool for the characterization of the liver metastatic T-cell landscape.

# Results

### Instrument calibration

The first step we undertook in the setup of the new workflow for HD flow cytometry was the instrument calibration. Using a BD FACS Symphony A5, we selected the appropriate photomultiplier voltage (PMTV) amplifications to generate high quality multicolor data. PMTV amplifications impact the resolution sensitivity, defined as the capacity to distinguish dimly stained elements from unstained ones (Hoffman, 2005; Perfetto et al, 2012). The stain index (SI) parameter is a commonly accepted indicator to evaluate the resolution of positive versus negative fluorescence signal (see the Materials and Methods section). To establish a suitable amplification to acquire human lymphocyte samples (Symphony A5 configuration shown in Table S1), we used PBMCs single stained with anti-human CD4 mAbs conjugated with 27 different fluorochromes emitting in all the detector channels. We then performed the process of PMTV titration (herein defined as "Voltration" and described in detail in the Materials and Methods section), which consists in evaluating the SI pattern of each CD4-stained sample as a function of progressively increasing PMTV in the corresponding channel (Figs 1A and S1). Ideally, the optimal voltages will correspond to the inflexion point of the curves. Amplifications below this point correspond to suboptimal resolution, whereas amplifications above this point can result in excessively bright signals. Once the preferred amplifications are selected, a visual inspection of the corresponding data plots verifies that negative signals are fully resolved above the bi-exponential scale, and positive signals do not exceed the detection range (Fig 1A). In the context of HD flow cytometry, fluorochromes with overlapping emission spectra cannot be avoided. This condition generates fluorescence signal spillover among neighboring channels with similar wavelengths even when they are excited by different lasers. A few fluorochromes (e.g., FITC) generate low or no spillover signal into other channels, whereas it is much more common for fluorescent dyes, particularly the tandem ones (e.g., PE-Cy5), to create significant spillover signal in neighboring detectors (Fig 1B) as a result of their overlapping spectral characteristics or the direct excitation of the acceptor dyes by the other laser sources. For this reason, we further evaluated the reciprocal spillovers to ensure that each

fluorochrome preferentially emits signal into its own detector. The end point of this calibration process is the setting of PMTV finely tuned to achieve maximal signal resolution (Table S1). To validate the cell-based Voltration procedure, we compared this approach with the bead-based Cyto-Cal/QCSB, which is well established as a reference protocol (Perfetto et al, 2012) (detailed in the Materials and Methods section, Fig S2 and Table S1).

### HD mAb panel design

To design the mAb panel to investigate our target T-cell populations by HD flow cytometry, we assigned antigens expressed at high and low densities to dull and bright fluorochromes, respectively (Maciorowski et al, 2017; Flores-Montero et al, 2019; Holmberg-Thyden et al, 2021), taking into account the reciprocal spread among dyes (Brummelman et al, 2019). The phenomenon of reciprocal spread reduces the SI that a given dye undergoes when its channel is impacted by a spread signal produced by the presence of a second dye, and it is typically a function of signal brightness and wavelength of affecting and affected dyes. Fig 1C exemplifies the impact of the PE-Cy5 generated spread on three different fluorochromes (BV421, APC-R700, and BB755), which consequently reduces the resolution of a double-positive population, while leaving the resolution of single-positive signals intact. This design approach resulted in a sensitive and reliable 26-color–based T-cell panel (Table S2) for HD flow cytometry to dissect both conventional and unconventional T cells infiltrating the liver metastases of CRC. To distinguish total leukocytes we included the CD45 antigen, together with CD3, CD4, and CD8 for the main conventional T-cell subsetting. To assess T-cell differentiation stages, CD45RA, CD62L, and CD95 antigens were chosen, allowing for the classification of CD45RA$^+$CD62L$^+$CD95$^-$ naïve, CD45RA$^+$CD62L$^+$CD95$^+$ memory stem, CD45RA$^-$CD62L$^+$ central memory, CD45RA$^-$CD62L$^-$ effector memory, and CD45RA$^+$CD62L$^-$ terminal effector T cells (Cieri et al, 2013; Gattinoni et al, 2017; Noviello et al, 2019). Moreover, the chosen antibody combination also allowed for the identification of T$_{FH}$ cells as CD45RA$^-$CD62L$^-$CD4$^+$CXCR5$^+$PD1$^+$ICOS$^+$ and Treg cells as CD4$^+$ CD25$^+$ CD127$^{low/neg}$. Unconventional T cells were unequivocally identified by staining with CD1d tetramer (tet), MR1tet, and anti-pan TCRγδ chain antibody, which bind the TCRs expressed by iNKT, MAIT, and γδT cells, respectively. Activation and functional exhaustion state of tumor-infiltrating T cells were investigated by detecting the expression of an extended range of immune receptors, such as: 2B4, LAG3, TIGIT, PD1, ICOS, OX40, GITR, CD39, HLA-DR, whereas CD69 and CD103 were used to label tissue-resident T cells (Dyck & Mills, 2017; Marin-Acevedo et al, 2018; Kim et al, 2021). Antibodies and tetramers were titrated to ensure an optimized staining (Fig S3 and Table S3).

### Performance evaluation of the Voltration instrument calibration applied to the HD mAb T-cell panel

We next validated the cell-based Voltration procedure with respect to the HD T-cell panel designed for the analysis of CRC-LM. First, we compared the Voltration calibration with the bead based Cyto-Cal/QCSB that is well established as a reference protocol (Perfetto et al, 2012) (see the Materials and Methods section, Fig S2 and Table S1).

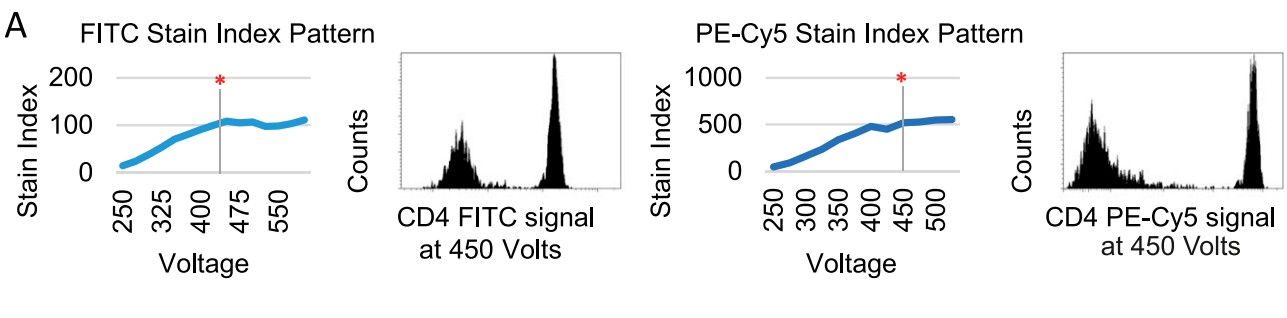

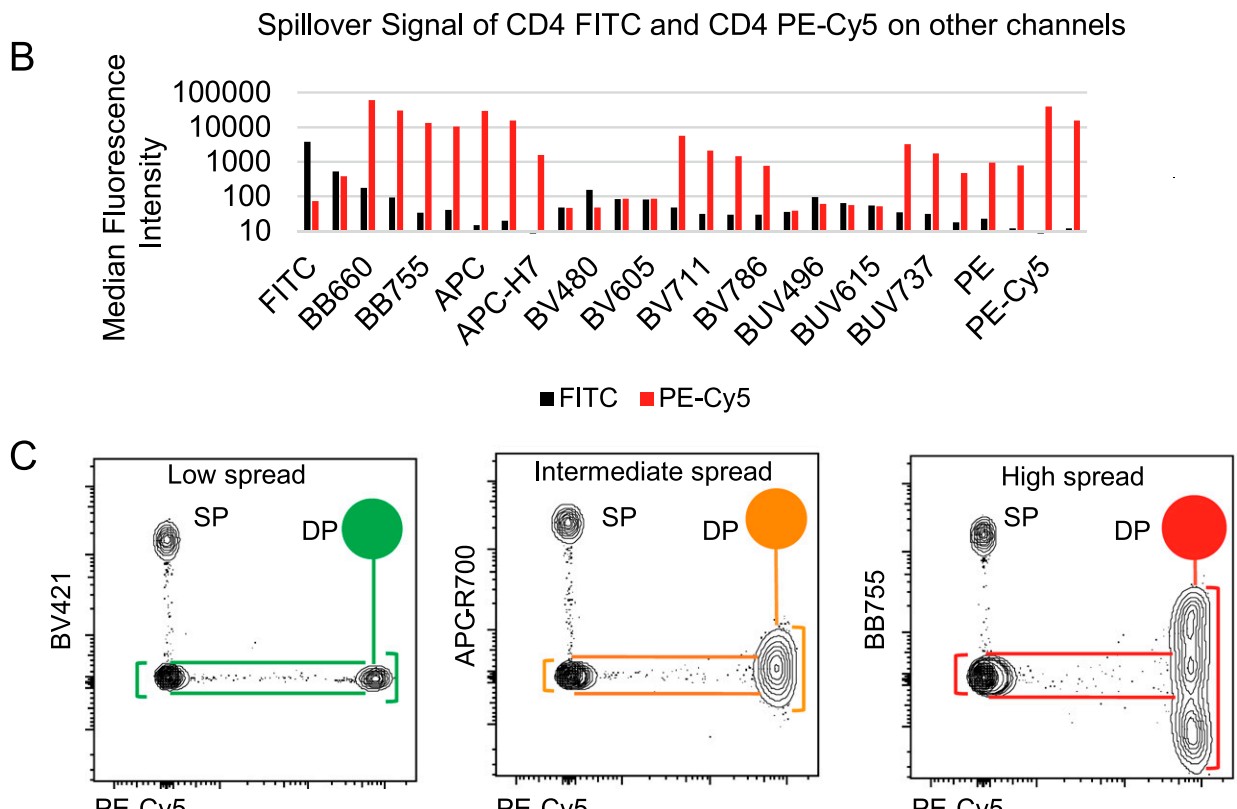

**Figure 1. PMTV setting by Voltration protocol.**
**(A)** The acquisition of single stained PBMCs was done at different voltages to calculate Stain Index Pattern as a function of PMT voltages and to determine the optimal setting. FITC and PE-Cy5 are shown as examples. Red asterisks and vertical bars represent the selected amplifications. Fluorescent signals at the chosen amplification were visually inspected on dot histograms to verify full resolution of negative signal and positive signal being on scale. **(B)** Fluorochrome signal spillover was checked on every detector to ensure that each color emits primarily into its specific channel. FITC is shown as an example of fluorochrome which is creating low spillover; PE-Cy5 is shown as an example of fluorochrome generating evident spillover signals in other channels. **(C)** Dot plot examples showing the impact of PE-Cy5 staining on BV421, APC-R700, and BB755 channels. Graphical representation helps visualize how the progressively increasing spread of PE-Cy5 into the three displayed channels reduces the resolution of double positive (DP), but not of single positive (SP) populations.

PBMCs were stained with the set of anti-human CD4 conjugated with 27 different fluorochromes described above and acquired with PMTV amplification selected by either Voltration or Cyto-Cal/QCSB calibration. The two approaches resulted in comparable patterns of stain indexes, although Voltration showed the tendency to generate higher signals, which were particularly evident in BUV395 and in the range of the Yellow Green excited fluorochromes (Fig S4A–C). Second, the two calibration procedures were compared by acquiring T cells stained with the 26 HD mAb T-cell panel, also

described above, to evaluate the impact of reciprocal interference among fluorochromes. For this specific purpose, T cells were polyclonally activated before staining by incubating PBMCs from healthy donors with PHA for 72 h to maximize the surface expression of the markers targeted by the mAb panel (Figs 2, S5A, and S6). Dot plot data analysis (Fig 2) showed a higher frequency of LAG3[+], TIGIT[+], GITR[+], and CD95[+] T cells detected using Voltration compared with the Cyto-Cal/QCSB setup procedure. Consequently, for example, a higher frequency of CD45RA[+]CD62L[+]CD95[+] memory

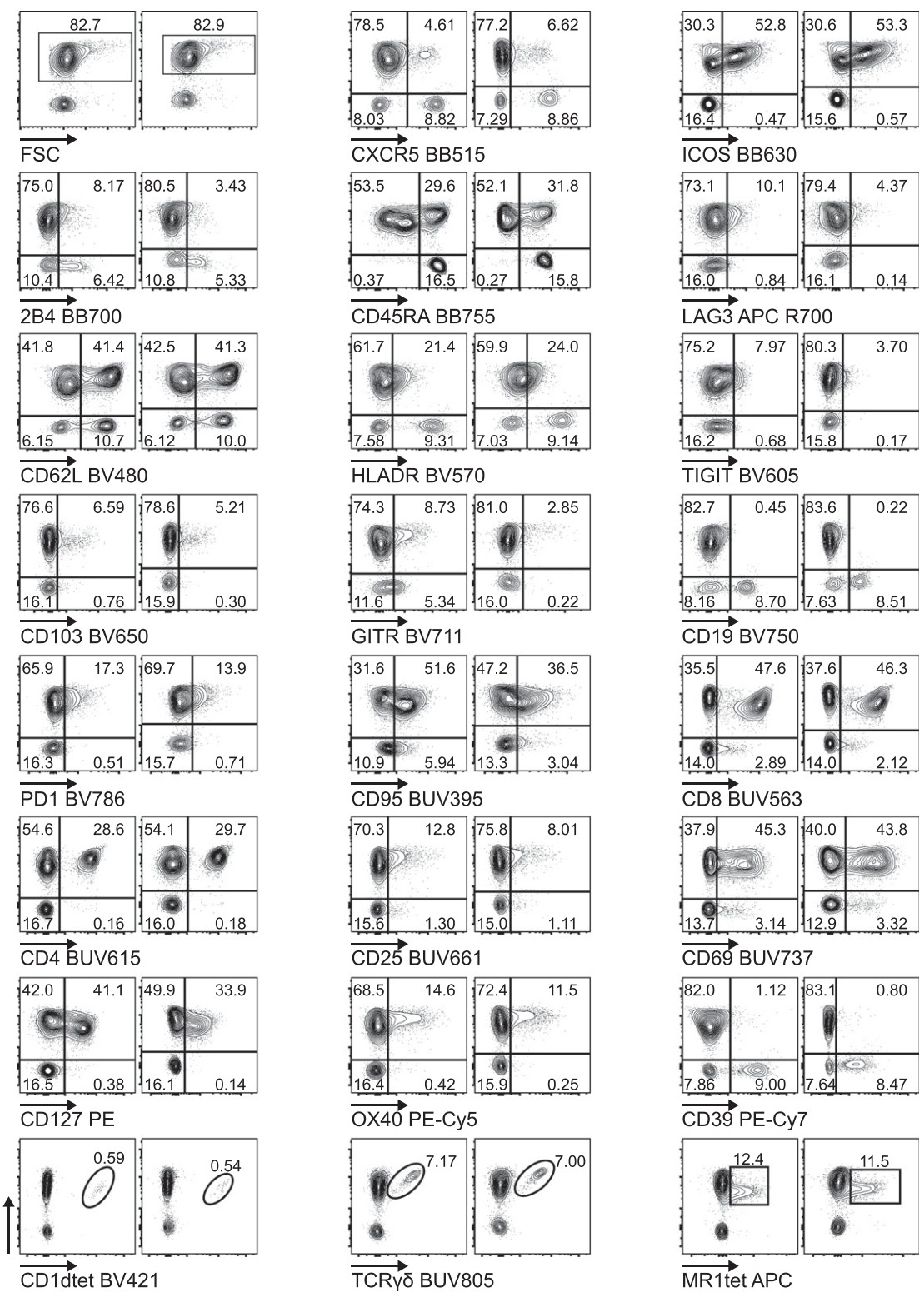

**Figure 2. Biological validation of the Voltration setting: Healthy PBMCs were purified by Ficoll density gradient and in vitro activated with PHA 1 µg/ml in complete medium culture.**

After 72 h, cells were harvested and stained with T-cell panel (Table S2 for antibody list and Fig S5A for gating strategy). Dot plot pairs for each conjugated mAb (x-axis) versus anti-CD3 (y-axis) are reported, with left plots resulting from Voltration and right plots from Cyto-Cal/QCSB calibration. C1dtet⁺, anti-pan-TCRγδ⁺, and MR1tet⁺ cells identify invariant natural killer T, γδT, and mucosal associated invariant T cells, respectively. Positivity was assessed keeping the CD3⁻ population or non-activated PBMCs from the same donor as the internal negative control. One representative experiment out of two is shown.

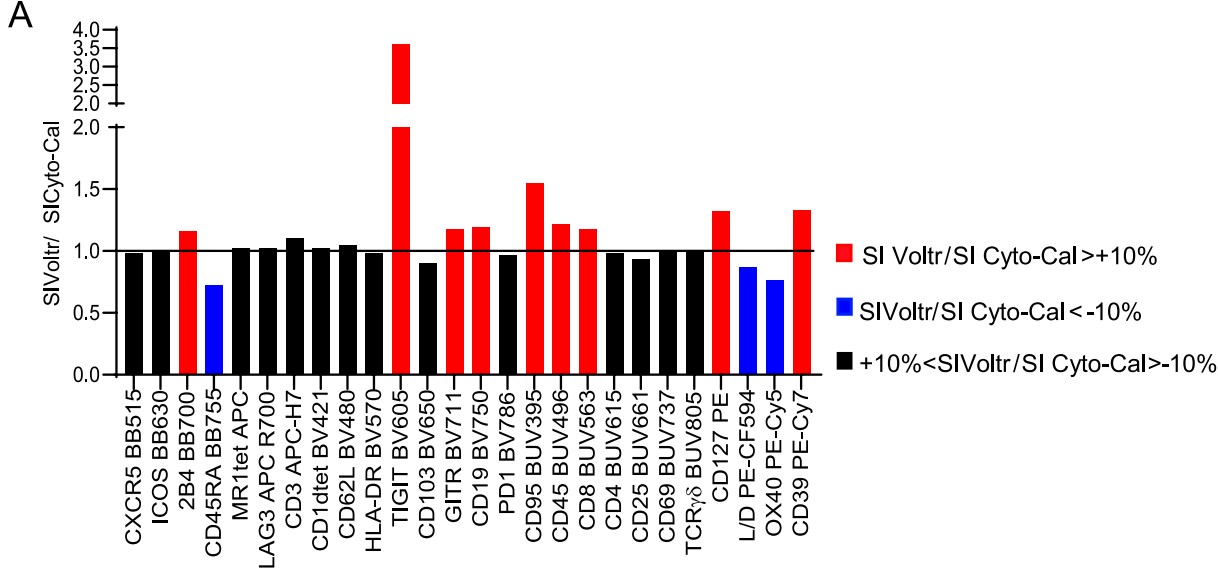

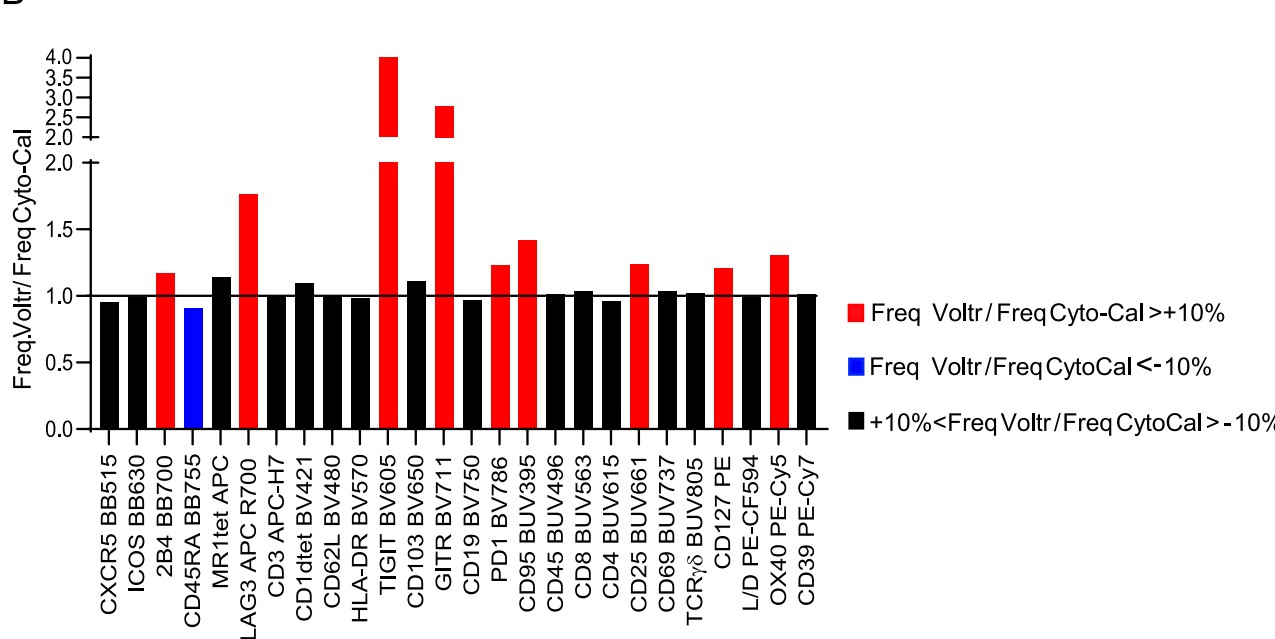

**Figure 3. Performance of Voltration setting.**
**(A, B)** Stain index (SI) (A) and positive cell frequency (B) ratios for each marker calculated with Voltration versus Cyto-Cal/QSCB settings (same staining as in Fig 2). Percentage differences >+10% or <−10% are represented by red bars and blue bars, respectively. All SI and frequency values were calculated within CD45⁺CD3⁺ T-cell population except for CXCR5, 2B4, HLA-DR, CD19, CD39 among CD45⁺CD3⁻ cells; CD3 among CD45⁺ cells; CD45 among FVS620⁻ live cells; and FVS620⁺ dead cells among singlet lymphocytes. One representative experiment out of two is shown.

stem cells could be identified on PBMCs cultured for 72 h in the absence of PHA and acquired with the Voltration setup than with Cyto-Cal/QCSB bead calibration, due to a lower resolution of the CD95⁺ T-cell population with the latter setting (Fig S5B). These observations were confirmed showing the ratios of SI (Fig 3A) or positive cell frequency (Fig 3B) calculated with the Voltration calibration versus those calculated with the Cyto-Cal/QCSB calibration. Overall, these results supported the use of the Voltration

calibration approach to set the PMTV for high quality multicolor flow cytometry with the designed HD T-cell panel.

### Tumor tissue processing

After setting up the staining conditions for the T cells, we investigated whether the processing conditions applied to obtain single cell suspensions from CRC-LM were affecting the optimal detection

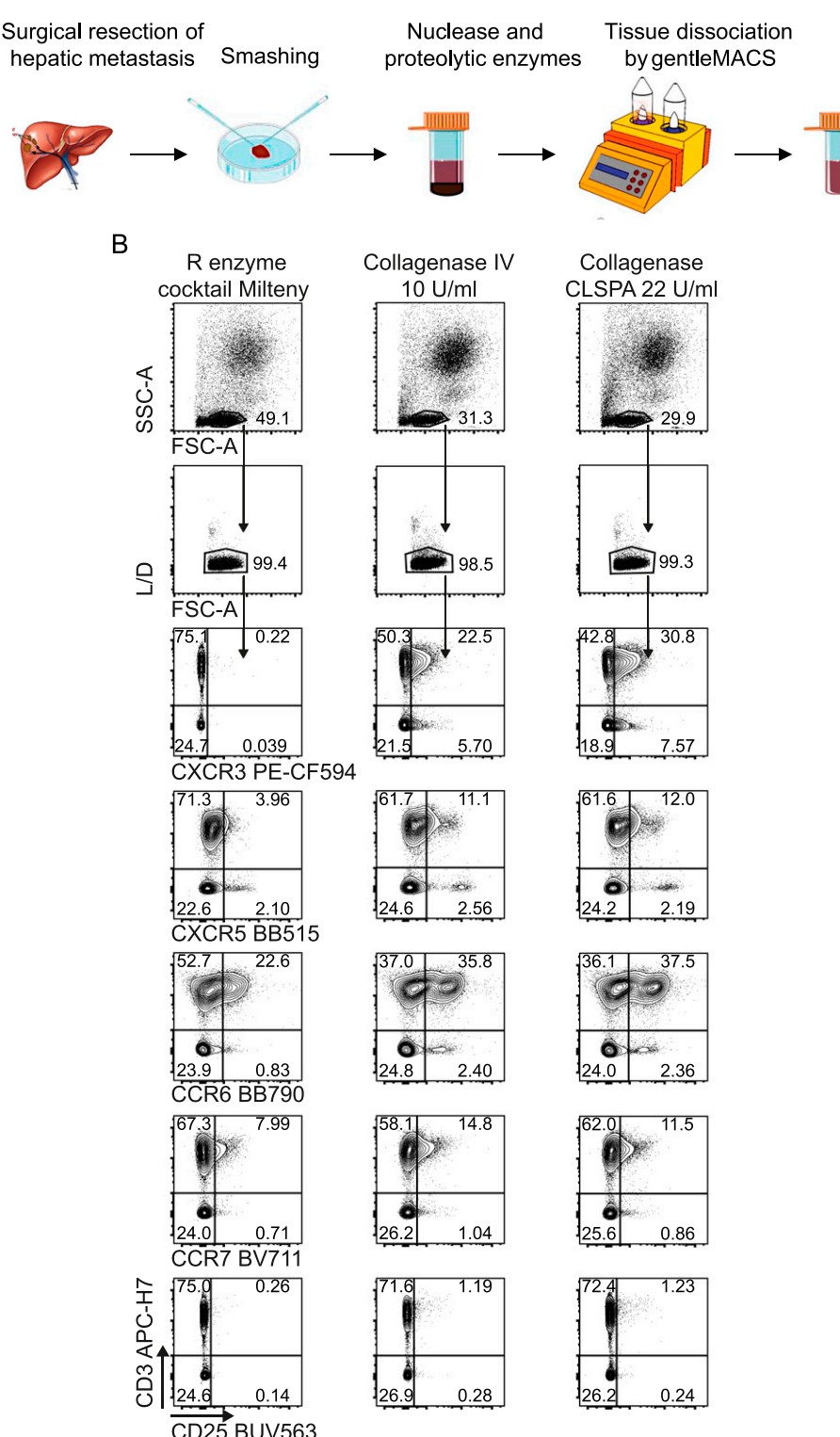

**A** Surgical resection of hepatic metastasis — Smashing — Nuclease and proteolytic enzymes — Tissue dissociation by gentleMACS

**Figure 4. Optimization of hepatic tissue processing to single cell suspension.**
**(A)** The metastasis was surgically resected, and a single cell suspension was obtained from a peritumoral tissue section as described. Different proteolytic enzymes (Enzyme R, Collagenase IV, or CLSPA Collagenase) were used together with Enzyme A and Enzyme H during the mechanical dissociation by gentleMACS. **(B)** The single cell suspension was stained and the expression of protease sensitive antigens (e.g., CXCR3, CXCR5, CCR6, CCR7, and CD25) by CD3[+] T cells was quantified by flow cytometry. One representative experiment out of two is shown.

of the targeted markers. Therefore, we focused on developing suitable protocols for the dissociation of surgically resected samples of CRC-LM to analyze their T-cell infiltrate. We combined an initial manual fragmentation together with a subsequent mechanical dissociation in the presence of enzymes for nucleic acid and proteolytic digestion (Figs 4A and S7). We selected the 37C_Multi_A_01 gentleMACS program for a 41-min dissociation during which the samples were incubated with nuclease and

proteolytic Enzyme A, Enzyme H, and Enzyme R cocktails according to the manufacturer's instructions (Fig 4A). However, this particular combination severely impacted the integrity of lymphocyte surface markers (Figs 4B and S7). To circumvent this drawback, we evaluated three different conditions of proteolytic treatment during the tissue dissociation: (i) fivefold reduced concentration of Enzyme R in the initially used volume; (ii) substitution of Enzyme R with chromatographically purified Collagenase CLSPA (22 U/ml) containing a high collagenase activity, but minimal contamination by proteolytic enzymes that could digest cell surface antigens; and (iii) substitution of Enzyme R with Collagenase IV (10 U/ml), a more crude extract commonly used to obtain single cell suspension from tissue for flow cytometry analysis (Trapecar et al, 2017). Slightly higher numbers of viable leukocytes were recovered with Enzyme R compared with CLSPA or Collagenase IV (Enzyme R yielded 0.3–1.8 × $10^6$, CLSPA 0.3–1.5 × $10^6$ and Collagenase IV 0.3–1.1 × $10^6$ cells/100 mg tissue from four different patients). We then verified the effects of each tissue processing protocol on the expression level of the markers known to be the most sensitive to proteolytic activity, and we observed that even a strongly reduced amount of Enzyme R completely abolished CXCR3 expression and lowered the staining intensity of CXCR5, CCR6, CCR7, and CD25, when compared with CLSPA and Collagenase IV. CLSPA better preserved both CXCR3 and CCR6 compared with Collagenase IV, which instead improved the resolution of only CCR7. CLSPA was therefore selected for the subsequent experiments. With the aim of comparing LM-CRC with primary colorectal cancer (CRC) immune infiltrates, we attempted to dissociate CRC samples. However, the abovementioned 41-min-long gentleMACS program did not lead to any recovery of leukocytes from primary CRC lesions. To retrieve viable leukocytes from CRC primary tumor (Range: 35,000–280,000 viable leukocytes), we prolonged the time of dissociation to 60 min (37C_h_TDK_1 program) along with treatment with Enzyme A, H, and CLSPA as for CRC-LM, improving the final quantitative and qualitative cell recovery.

### Phenotypic characterization of tumor-infiltrating lymphocytes

We applied the optimized methodology described above to stain fresh single cell suspensions derived from CRC-LM and distal normal liver (NL) samples, obtained from the same patient (Figs 5, S8A, and S9–S11). The expression of CXCR5, ICOS, HLA-DR, TIGIT, CD103, PD1, CD95, CD25, CD69, and CD39 was clearly higher among tumor-infiltrating T cells than within normal hepatic tissue. We also noticed that 2B4, previously reported to be associated with T-cell exhaustion in the TME, was expressed at higher level by CD3$^+$ and/or CD3$^-$ cells from the normal tissues than the tumor tissues. In addition, $\gamma\delta$T, MAIT, and iNKT cells could be detected both in normal liver and neoplastic tissue samples. Moreover, rare intratumoral CXCR5$^+$PD1$^+$ICOS$^+$ T$_{FH}$ and CD4$^+$CD25$^+$CD127$^-$ Treg cells were more frequent in the tumor than the normal liver tissue (Fig 6A and B). We also applied the aforementioned T-cell antibody panel to stain single cell suspensions from primary CRC tissue sections, comparing phenotypes of circulating and tumor-infiltrating T cells from the same patient (Figs 7, S8B, S12, and S13). The expression of LAG3 and CD62L, which were almost undetectable in hepatic tissue, was evident in primary CRC infiltrating and circulating T cells, respectively, hence excluding the possibility that these antigens could

have been damaged by the tissue processing procedure or that their specific antibodies were not properly titrated. We also assessed the performance of the T-cell mAb panel on LM-CRC single cell suspensions that had been previously frozen. However, we observed that the expression level of some exhaustion markers, mostly LAG3 and PD1 on T cells, and the frequency of unconventional T cells were lower in thawed samples in comparison with fresh samples (Fig S14). Collectively, these observations confirmed that the described procedures facilitate the accurate detection of phenotypic profiles of T cells infiltrating CRC-LM, which was optimal for freshly drawn and serially acquired consecutive samples.

### Identification of tumor-infiltrating unconventional T cells by computational analysis

We challenged the accuracy and sensitivity of our workflow by verifying whether also less characterized tumor-infiltrating unconventional T-cell populations could be identified by unsupervised computational analysis, in addition to classical analysis approaches. Unsupervised analysis of the previously described CRC-LM infiltrating T-cell data set (Fig 5) was carried out by cytoChain, a recently published web application for HD flow cytometry data mining (Manfredi et al, 2021). The cytoChain modular pipeline includes pre-analytical steps to correct flow cytometer fluctuations and multidimensional data scattering; evaluation of the appropriate HD analysis according to the specific data set qualities; and quantitative analysis to identify clusters of cells sharing similar phenotypes with an exhaustive graphical output. The first pre-analytical step was to exclude the rare fluorescence signal fluctuations caused by unstable instrumental acquisition by FlowAI (Monaco et al, 2016). Second, to mitigate the complexity of the flow cytometry data distribution resulting from the variance of different fluorescence intensities (Finak et al, 2010), the cleaned data were transformed by arcSinh scaling. Finally, a density correction was performed by Spanning-tree Progression Analysis of Density-Normalized Events (SPADE) (Qiu et al, 2011) to isolate out the multidimensional scattered outliers (3% events were censored and 22,727 cells were analyzed) (Fig S15A). In agreement with our previous report, this data optimization did not significantly affect the original frequency of rare cells such as iNKT, $\gamma\delta$T, and MAIT cells nor the distribution of these cells in subsets based on CD4 and CD8 expression (Fig S15B and C). We then performed the dimensionality reduction of the optimized data set applying the t-distributed Stochastic Neighbor Embedding (t-SNE) algorithm (Mahfouz et al, 2015), and the expression level of each marker within the whole T-cell population was visualized by t-SNE heat-maps (Fig 8A). As expected, large areas of the t-SNE heat maps were dominated by CD4$^+$ and CD8$^+$ T cells. Nonetheless, CD1dtet$^+$, MR1tet$^+$, and anti-pan TCR$\gamma\delta^+$ cells could be distinguished despite their low frequency (Figs 8A and S16A). To further dissect the data set, we applied the FlowSOM algorithm (Van Gassen et al, 2015), which could not selectively assign iNKT cells to any cluster or meta-cluster even though they were tightly grouped on the t-SNE map (Fig S16A and B). To overcome these problems, we implemented our recently described cytoChain application (Manfredi et al, 2021) with FastPhenoGraph (FastPG) algorithm (Fig S16C) (Bodenheimer et al, 2020 Preprint), which computed 28 clusters that were superimposed on

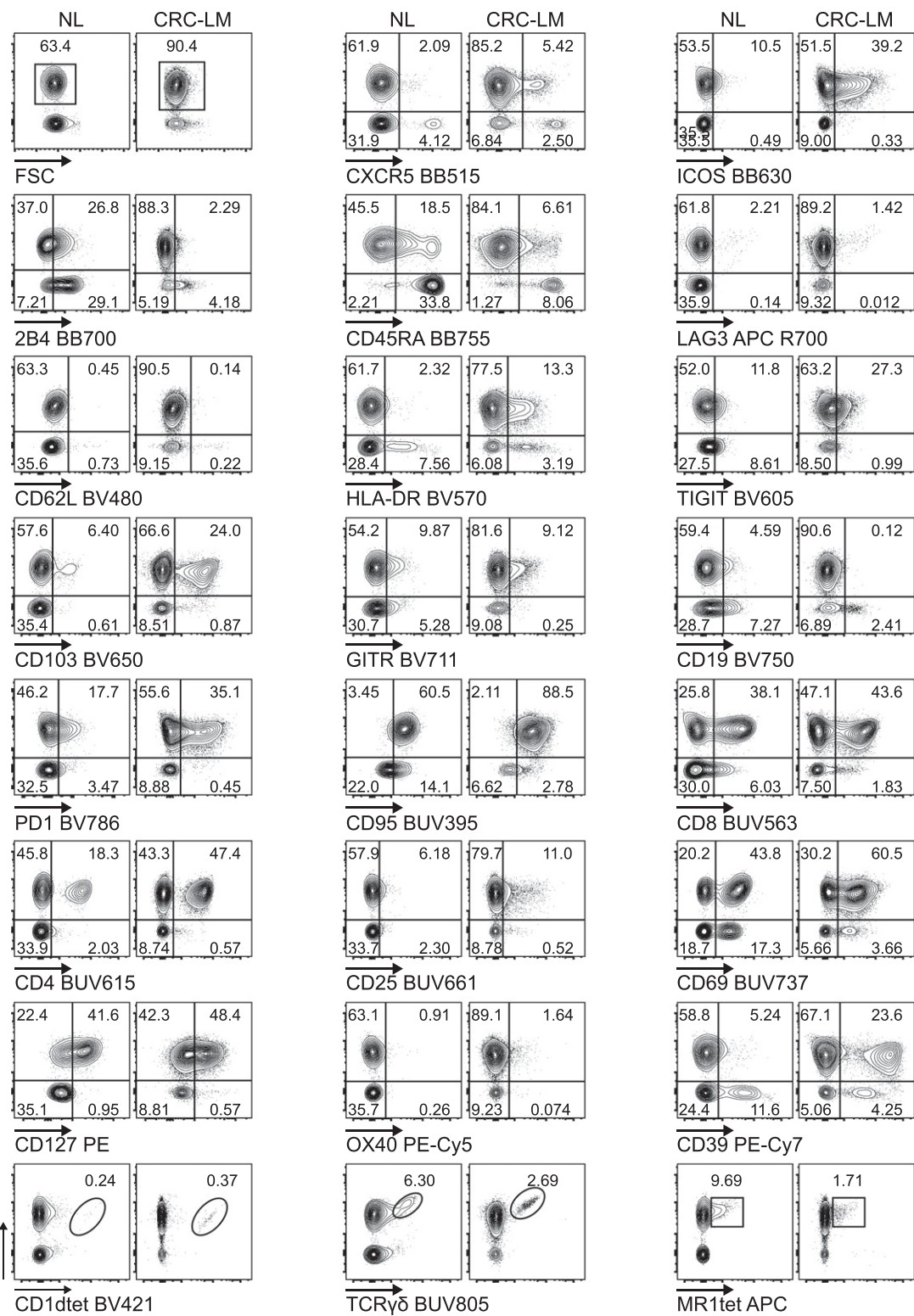

**Figure 5. Performance of Voltration instrument setting associated with hepatic tissue processing and analysis of T-cell infiltrate from NL and CRC-LM tissue sections.**
NL and CRC-LM samples from the same patient were freshly processed by mechanical and enzymatic dissociation with gentleMACS + Enzyme A + Enzyme H + CLSPA. Cells were harvested and immediately stained with T-cell panel (Table S2 and gating strategy Fig S8A). Dot plots pairs for each conjugated mAb (x-axis) versus anti-CD3 antibody (y-axis) refer to lymphocytes recovered from NL and CRC-LM, as indicated. CD1dtet⁺, anti-pan-TCRγδ⁺, and MR1tet⁺ cells identify unconventional T-cell populations of invariant natural killer T, γδT, and mucosal associated invariant T cells, respectively. One representative patient out of four is shown.

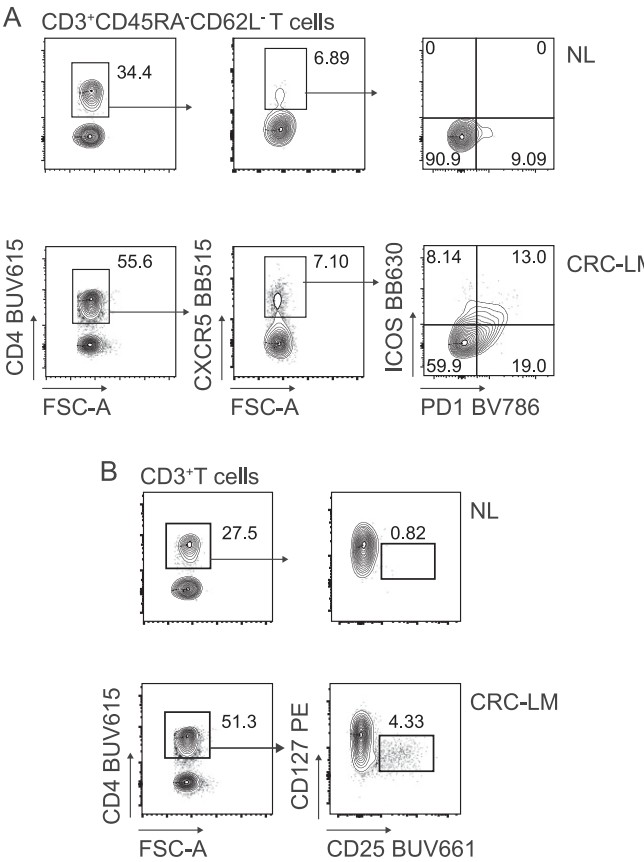

**Figure 6. Identification of T_FH and Treg cells infiltrating NL and CRC-LM tissues.**
**(A)** T_FH cells infiltrating NL and CRC-LM (Fig 5) tissue as indicated: total T_FH cells were identified as memory CD45RA⁻CD62L⁻ CD4⁺ CXCR5⁺ T cells, among which the activated T_FH subset was PD1⁺ICOS⁺. **(B)** Putative Treg cells infiltrating NL and CRC-LM were identified as CD4⁺CD127^low/neg CD25⁺ T cells.

the t-SNE map, allowing both the visualization of their distribution within the data set (Fig 8B) and of their phenotype by a heat-map generated based on the fluorescence intensity associated to each expressed marker (Fig 8C). For instance, a specific phenotype signature was found to be enriched among CD8 T cells for clusters 21, 17, 24, 13, 27, and 23 that expressed both tissue residence (CD69-CD103) and activation/exhaustion markers to a variable extent (i.e., CD39, CD95, TIGIT, 2B4, PD1, HLA-DR, ICOS, and GITR). Accordingly, the cell frequency within these clusters was higher among CD8⁺, than CD4⁺ and CD4⁻CD8⁻ (double-negative DN) cell subsets (Fig 8D–G). To verify whether rare T-cell subsets could be comprehensively retrieved by unsupervised analysis with cytoChain, a deeper analysis was performed. Manually gated iNKT, MAIT, and γδT cells from the optimized data set (Fig 9A and B) and the location occupied by CD1dtet⁺, MR1tet⁺, and anti-panTCRγδ⁺cells on tSNE maps (Fig S16A) are shown. Among the 28 clusters computed by FastPG, the algorithm captured clusters 1 and 2 that homogeneously contained CD1dtet⁺ and anti-pan TCRγδ⁺ cell populations, respectively. Unlike CD1dtet⁺ and anti-pan TCRγδ⁺ cells, MR1tet⁺ cells were fragmented in clusters 3–6 (Fig 9C). The CD8⁺, CD4⁻CD8⁻, and CD4⁺ subset composition revealed that the CD1dtet⁺, anti-pan TCRγδ⁺, and MR1tet⁺ cells within clusters 1–6 (Fig 9C) resembled those found for

manually gated iNKT, TCRγδ, and MAIT cells (Fig 9A). Moreover, the heterogeneity among MAIT cells, which was responsible for their fragmentation in different clusters, derived not only from a differential expression of CD4 and CD8 markers, but also from different levels of fluorescence intensity associated to MR1 tetramer binding, with the lowest level shown by the fraction of MAIT cells contained in cluster 6 (Fig 9C). Clusters 1–6 were overlaid on t-SNE maps (Fig 9D) and their positions corresponded to those found for CD1dtet⁺, MR1tet⁺, and anti-panTCRγδ⁺ cells on tSNE maps Fig S16A, further confirming the cell identity of the unconventional T-cell subsets identified by the computational unsupervised analysis. An even greater complexity was revealed by cluster phenotype deconvolution (Fig 9E), which showed a differential expression of markers despite the paucity of cells contained within the clusters. For instance, although iNKT cells in cluster 1 and MAIT cells in cluster 6 were both mainly CD4⁺, they displayed distinct phenotypes with a higher expression of CD127 in cluster 1 and of CD95, ICOS, CD25, HLA-DR, TIGIT, OX40, and CD39 in cluster 6. Together, these results indicate that the workflow is appropriate for unsupervised HD analysis and can accurately phenotypically dissect even rare populations of tumor-infiltrating T cells (Saeys et al, 2016; Liechti et al, 2021).

## Discussion

We have defined a methodological workflow that allows the precise characterization of the T-cell landscape of CRC liver metastases by HD flow cytometry. Each single step of the process was carefully implemented: (1) validation of a new cell-based flow cytometer PMTV setting, (2) design of a 26-color panel for T-cell landscape definition that includes both antibodies and tetramers together with an appropriate staining protocol, (3) optimization of tissue mechanical and enzymatic digestion conditions for single cell suspension, and (4) computational analysis of the flow cytometry data by a recently published application specific for the capture of very rare cell subsets.

A fundamental prerequisite for high signal resolution in HD flow cytometry is the appropriate instrumental setting. Two different definitions of "sensitivity" exist in flow cytometry: the first is known as *threshold sensitivity* and is defined as the lowest light signal that can be discriminated from background. The second is named *resolution sensitivity* and refers to the capacity of discriminating dimly stained from unstained particles (Wang & Hoffman, 2017). Notably, whereas *threshold sensitivity* is a pure function of instrument design and characteristics, *resolution sensitivity* heavily relies on the appropriate setting of PMTVs, thus highlighting how critical this process is for generating high quality multicolor data. The most widely accepted workflow to define the appropriate instrument setting is an elegant Cyto-Cal/QCSB bead based procedure (Perfetto et al, 2012). Here, we used this well-established approach as a reference starting point to evaluate a similar process for PMTV selection (Voltration), which was based on cells rather than beads. As beads are a synthetic sample, they are less sensitive to handling and relatively free of debris particles; thus, they have the potential to facilitate gating strategies and the generation of reproducible data to guide instrument setup. On the other hand,

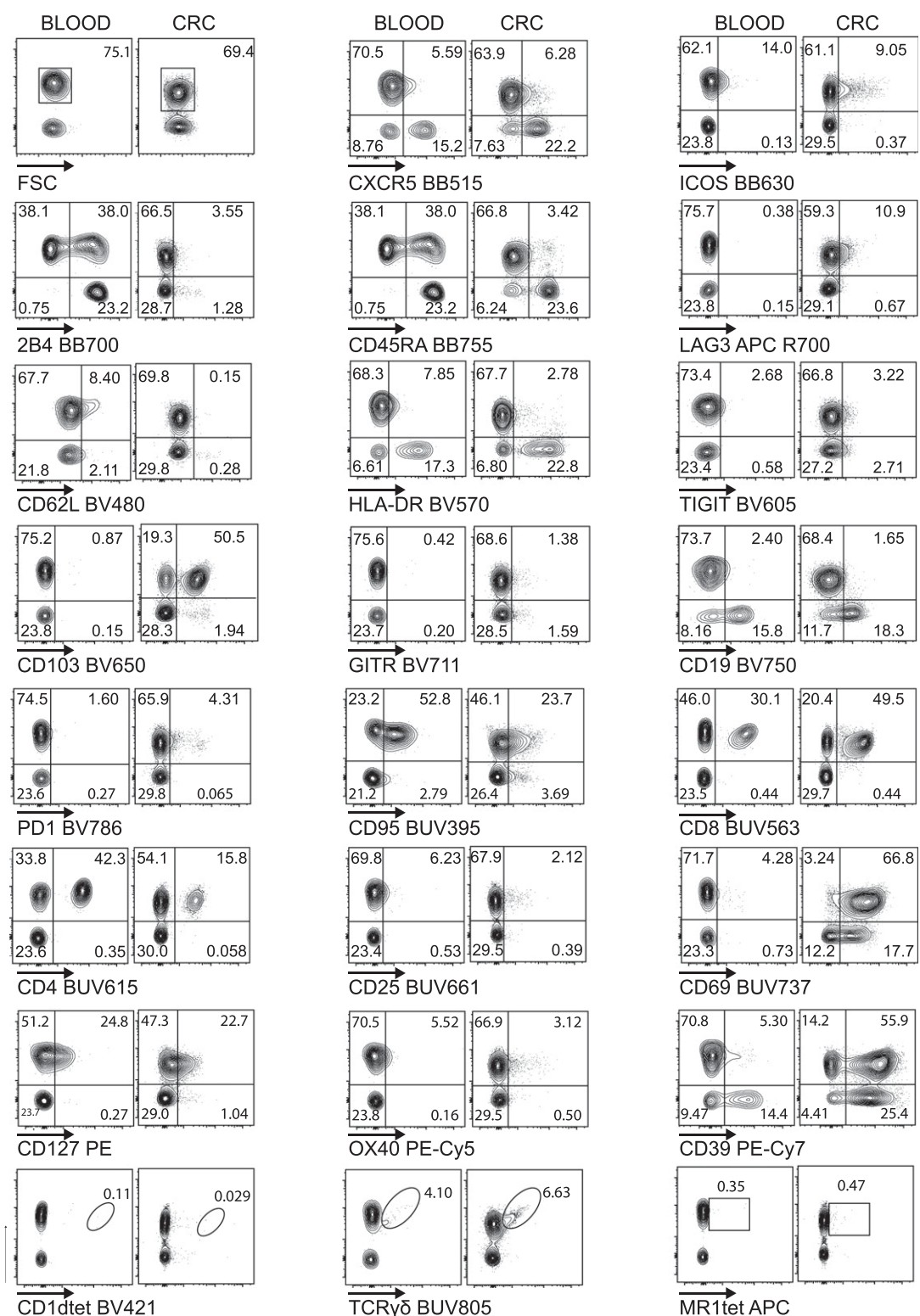

**Figure 7. Performance of Voltration instrument setting on primary CRC infiltrating T cells.**
CRC tissue was processed by mechanical and enzymatic dissociation with gentleMACS + Enzyme A + Enzyme H + CLSPA. CRC infiltrating and peripheral lymphocytes from pre-surgical whole blood from the same patient were stained with the T-cell panel (Table S2 and gating strategy Fig S8B). Dot plot pairs for each conjugated mAb (x-axis) versus anti-CD3 antibody (y-axis) compare pre-surgical whole blood lymphocytes and CRC infiltrating lymphocytes as indicated. CD1dtet[+], anti-pan-TCRγδ[+], and MR1tet[+] cells identify invariant natural killer T, γδT, and mucosal associated invariant T cells, respectively. One representative patient out of three is shown.

**Figure 8. Unsupervised computational high dimensional analysis of LM-CRC infiltrating T cells.**
Flow cytometry data from CRC-LM infiltrating T cells (see Fig 5) were cleaned by FlowAI, scaled by arcSinh, and corrected by SPADE test. **(A)** t-SNE heat-map for each marker was applied on 22727 CD45⁺CD3⁺ events. **(B)** FastPG algorithm computed 28 clusters that were superimposed on t-SNE map and identified by numbers and colors as indicated on the right. **(C)** Heat-map represents the marker fluorescence intensity associated to each cluster. **(D, E, F)** Cluster distribution on CD4⁺, CD8⁺ and DN t-SNE maps. **(G)** Cell frequency within each cluster among CD4⁺, CD8⁺, and DN T cells.

beads can have different optical characteristics from cells in certain fluorescent channels, and are therefore not representative of the biological samples to be analyzed. Although cells (PBMCs in our case) represent biological samples better, they are much more sensitive to processing, prone to quality issues, and more difficult to standardize. Voltration protocol relies heavily on the concomitant visual inspection of both SI values versus PMTV value graphs

and the emission pattern of each fluorochrome in all the detection channels, allowing the fine tuning of PMTV values for optimal signal resolution and minimal spillover. Overall, the two approaches performed similarly, although in the context of the whole T-cell panel, Voltration allowed for better resolution of dim signals associated to exhaustion markers such as LAG3, TIGIT, and GITR, which are a crucial topic of our study of CRC-LM.

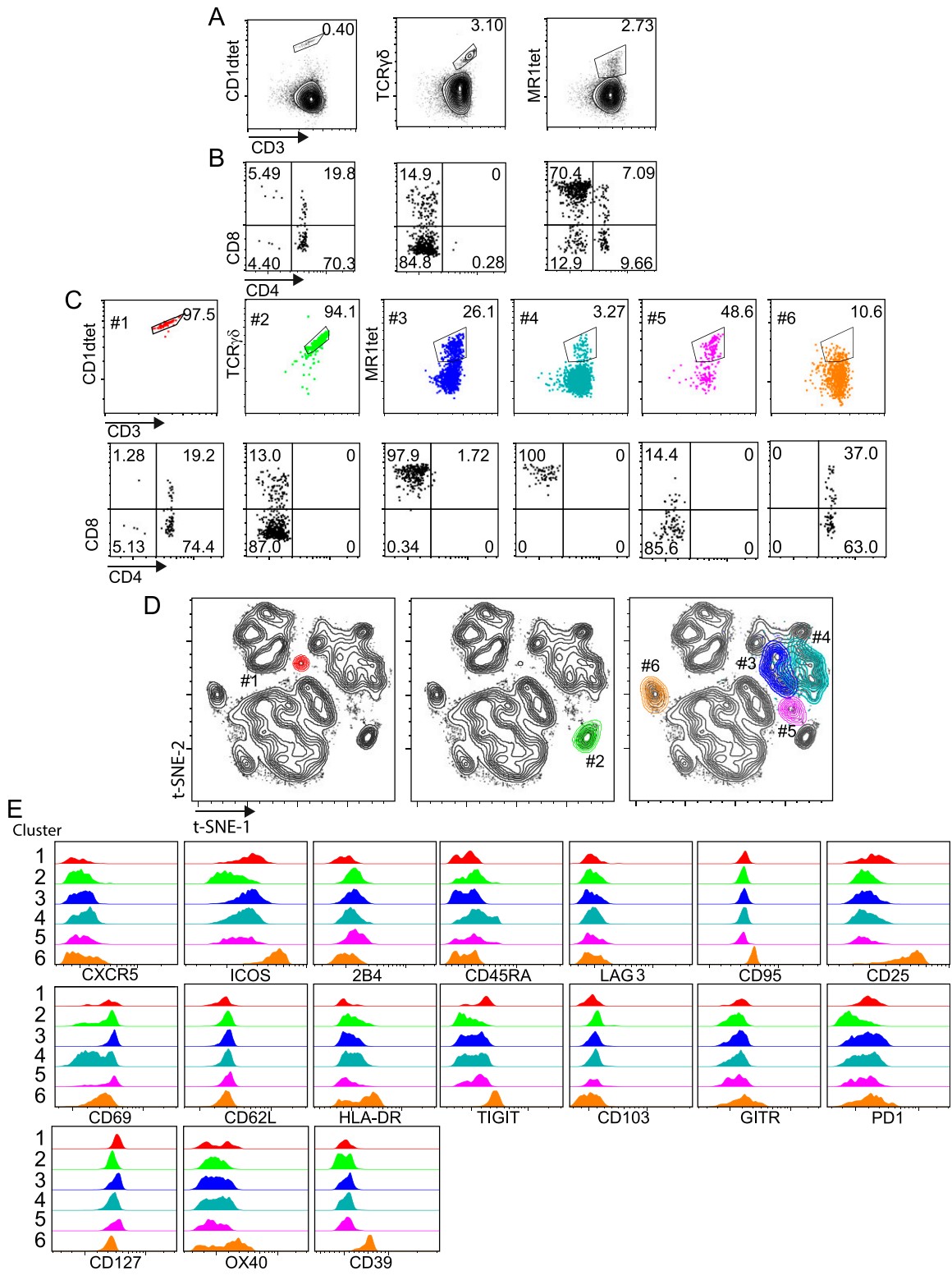

**Figure 9. CRC-LM infiltrating unconventional T cells are captured by unsupervised computational high dimensional analysis.**

Pre-analytical steps were applied on infiltrating CRC-LM T cells (see Fig 5) data set as in Fig 8. **(A, B)** The frequency of invariant natural killer T (iNKT), γδT, and mucosal associated invariant T (MAIT) cells among total T cells and their distribution in CD4⁺, CD8⁺, CD4⁻CD8⁻, and CD4⁺CD8⁺ subsets obtained after optimization are indicated (frequencies registered before optimization are shown in Fig S15B). **(C)** FastPG algorithm captured 28 clusters among which cluster #1 and #2, respectively, contain CD1dtet⁺ and anti-pan γδ TCR⁺ cells, whereas clusters #3 to #6 are enriched in MR1tet⁺. The frequency of iNKT, γδT, and MAIT cells within each cluster, together with their frequency within subsets based on CD4 and CD8 expression are shown. **(D)** Clusters #1 to #6 were manually overlaid on t-SNE maps as indicated. **(E)** Phenotype de-convolution of clusters in D is reported for each marker.

Indeed, the optimal resolution of the panel was achieved by appropriate assignment of dyes to specific antigens and careful mapping of spread signals.

The processing of highly heterogeneous tissue sections from colon and liver samples into a single cell suspension was also addressed. In the present work, we report optimal mechanical dissociation conditions for CRC and LM-CRC with gentleMACS equipment that have not been previously described. After finding that the enzymatic cocktail proposed by the gentleMACS equipment manufacturer was too aggressive, we selected chromatographically purified collagenase CLSPA to facilitate tissue disaggregation while preserving the integrity of the surface antigens considered most sensitive to proteolytic treatment, such as certain chemokines (e.g., CXCR5, CXCR3, and CCR6) or cytokine receptors (e.g., CD25 IL-2Ralpha).

To validate our new workflow, we processed normal liver and CRC-LM tissues; stained the obtained single cell suspensions with the 26-color T-cell panel; and acquired them with Symphony A5 calibrated by Voltration procedure. Our workflow yielded excellent detection resolution for tumor-infiltrating exhausted T-cell subsets in comparison with normal liver tissues along with the identification of $\gamma\delta$T and MAIT cells and even of very rare iNKT cells. Deep resolution of the T-cell landscape revealed differential distribution of $T_{FH}$ subsets and Treg frequencies among T cells infiltrating normal and neoplastic lesions. Similar sensitivity was also obtained with the primary CRC tissue section and whole blood samples, confirming the possibility to compare peripheral blood, primary lesion, and derived metastasis concomitantly obtained from the same individual for the description of the whole T-cell landscape in different compartments. Furthermore, we observed that the freezing and thawing procedure negatively impacted certain antigens' integrity and rare T-cell population frequencies, implying that our optimized workflow best relies on the serial acquisition and staining of fresh cell suspensions obtained from consecutive surgical metastatic resections. This strategy requires a continuous quality control to standardize the instrument and reagents over time, which involves the use of CS&T and rainbow beads to detect laser and/or fluidic chamber failures and to standardize the acquisition procedure as well as the titration of each new batch of reagents (i.e., antibody, cell vitality dye, and tetramers) to find the optimal concentration for the best staining resolution (Cossarizza et al, 2019).

Finally, we demonstrated that our workflow was sensitive enough to permit the identification of rare populations by unsupervised computational analysis. For example, the relatively rare unconventional T-cell populations, that is, iNKT, MAIT and $\gamma\delta$T cells, were clearly identified by cytoChain (Manfredi et al, 2021), our recently published computational workflow that integrates several different available HD analytical tools. The FlowSOM algorithm, although extremely efficient for several applications, did not succeed in capturing iNKT cells in a single cluster. Meanwhile, the FastPG algorithm, captured both iNKT cells and $\gamma\delta$T cells within two distinct clusters.

The purpose of this study was to set up a workflow able to assess also total $\gamma\delta$ T cells from liver metastases, but not yet meant to dissect the $\gamma\delta$ TCR repertoire composition. Among $\gamma\delta$T cells, the V$\delta$1$^+$ subset is known to enrich in human liver (Hunter et al, 2018),

although a sizeable fraction of the V$\gamma$9V$\delta$2 can also be detected (Zakeri et al, 2022). V$\delta$1$^+$ T cells can directly kill primary CRC cells, but they seem also to play a pro-tumorigenic role (Suzuki et al, 2020). However, the specific functional phenotype and TCR repertoire of the $\gamma\delta$ T-cell subsets in LM-CRC are ill defined and we are currently gaining further insight into this issue in a large cohort of prospectively recruited patients. Furthermore, FastPG sub-fractioned MAIT cells in 4 clusters based on their CD4 and CD8 expression, with CD4$^+$ cells exhibiting the lowest fluorescence intensity associated with MR1tet. This MAIT cell heterogeneity was also found on the original data set by manual gating, demonstrating that it was not an artefactual result of the computational analysis. We could also further dissect MAIT cells by phenotype de-convolution of each cluster and found an interesting cluster expressing a distinctive CD4$^+$CD95$^{hi}$CD25$^{hi}$ICOS$^{hi}$HLA-DR$^{hi}$TIGIT$^{hi}$OX40$^{hi}$CD39$^{hi}$CD127$^{low}$ phenotype. Relatively infrequent CD4$^+$ MAIT cells with a similar phenotype were previously described in the peripheral blood from healthy donors (Gherardin et al, 2018) and in primary CRC tissue infiltrate (Li et al, 2020), further confirming the reliability of our approach.

Collectively, we describe a novel workflow, permitting an in-depth definition of conventional and unconventional T-cell functional status in CRC-LM in large cohorts of patients. This workflow can be adapted both to investigate other tissues and to assess other immune cells known to play a critical role in the TME, such as myelo-monocytic and B cell populations, to define their interplay with T cells.

# Materials and Methods

### Symphony A5 calibration by Voltration protocol

The instrument setting procedure and sample acquisitions were performed with BD FACSymphony A5 cell analyzer (BD Biosciences) located in the ISO 9001 certified Flow Cytometry facility FRACTAL at Ospedale San Raffaele Scientific Institute. The Voltration protocol was based on PBMCs single stained with anti-human CD4 (clone SK3) conjugated with 27 different fluorochromes (BD 566352 Human CD4 Fluorochrome Evaluation kit; BD 624371 Human CD4 Prototype Fluorochrome Evaluation kit). To ensure optimal staining, CD4 antibodies conjugated with primary and last fluorochrome for each laser line (BUV395-BUV805 Laser UV; BV421-BV786 Laser Violet; BB515-BB700 Laser Blue; PE-PE-Cy7 Laser Yellow Green; APC-APC-H7 Red) were pre-titrated (8 points) and optimal concentration, resulting in 0.8 $\mu$g/500,000 PBMCs, was visually determined to obtain the highest positive/negative signal ratio. This amount of reagent was adopted to stain all 27 samples. Staining was carried out for 15 min at r.t., with 500,000 PBMCs/sample resuspended in 100 $\mu$l of PBS supplemented with 10% Normal Human Serum (NHS) (Euroclone Cat. no. ECS0219D). After incubation cells were washed three times at 510$g$ for 5 min in 200 $\mu$l PBS, resuspended in 200 $\mu$l PBS, and immediately sent for acquisition.

To generate the Voltration setting, BD FACSDiva was set creating one experiment for each laser line. Within the experiment, specimens had the names of tested fluorochromes, and tubes the names of the adopted voltages. Minimum voltage was set at 250 and

increased by 25 V up to a maximum of 750 V unless the signal was out of scale with the lower amplification. Samples were acquired using flow rate low collecting 1,500 lymphocytes/tube in the absence of fluorescence compensation.

After acquisition, the Stain Index $\frac{(Median_{Positive} - Median_{Negative})}{2 \times rSD_{Negative}}$ (SI) was calculated for each tube and plotted as a function of voltage for every fluorochrome. The optimal voltage for each fluorochrome was visually selected based on the following criteria:

(i) At low amplification SI grows rapidly with the increase of PMTVs. At optimal amplification voltage, the SI curves normally show an inflection point, from which further increase of the SI is greatly reduced. Amplifications below the inflection points correspond to suboptimal resolution, whereas amplifications above the inflection point result in a moderate gain of resolution and often in the generation of excessively bright signals.
(ii) Achieve full resolution of the negative signal.
(iii) Place positive signal within linear range of amplification.
(iv) Each fluorochrome emitting primarily in its detector and to a less or similar extent into other detectors.
(v) In case no inflection point was evident for a given detector, voltage was set in range with other detectors of the same laser, reading dyes at closer wavelengths.

Having ensured that all criteria were met, a compensation matrix was generated using freshly stained PBMCs in a dedicated experiment.

Most of the compensation resulted below 100%, the highest being BV480–BUV496 = 123%.

## Symphony A5 calibration by Cyto-Cal/QSCB based protocol

The following reagents were prepared:

(1) Cyto-Cal beads containing broad spectrum dyes at five different concentrations and one negative peak (one drop corresponding to 20 $\mu$l of particles into 1 ml of PBS in a 12 × 75 mm tube), rainbow calibration particles (8 peaks, 3.0–3.4 $\mu$m) (one drop corresponding to 20 $\mu$l of particles into 1 ml PBS) used for a better resolution of UV-395 fluorescence peaks.
(2) Unstained COMP beads (40 $\mu$l of particles into 60 $\mu$l PBS) to finely locate the negative background in case it is not resolved by Cyto-Cal reagent.

(i) Each of the above reagents was separately acquired from 250 to 800 V with increasing gains of 50. The fluorescence signals of Cyto-Cal beads were plotted versus the voltage values for each detector and the Gain Range Calibration Tool was outlined on the plot by two boxes: box 1 ranges from the negative background to the first positive peak (separation distance) and box 2 from the first to the second positive peak (linearity distance) (Fig S2A). The voltage is chosen within a range of optimal sensitivity (box 1, separation distance, maximal) and linearity (box 2, linearity distance, constant).
(ii) The selected voltage was validated with capturing antibodies quantum simply cellular beads (QSBCs) (Fig S2B) with

1 negative peak (M1) and four peaks with increasing intensities (from M2 to M5). The beads were stained with 0.8 $\mu$l/sample anti-human CD4 conjugated with 27 different fluorochromes (BD 566352 Human CD4 Fluorochrome Evaluation kit; BD 624371 Human CD4 Prototype Fluorochrome Evaluation kit) and acquired in the range of the selected voltage increased or decreased by 25 V for each detector and M2 and M5 ratios were calculated as: M2 ratio = MFI of the lowest positive peak/90th percentile of the negative bead, and the M5 ratio = MFI of the highest positive peak/90th percentile of the fluorescence signal from the negative bead negative (Fig S2C). The highest M2 and M5 ratio defining the separation of dimmest and brightest peaks, respectively, from the negative peak allowed us to determine the optimal voltage for each detector.

## Antibody panel design criteria

A multicolor panel design has been performed based on consolidated flow cytometry best practice.

Spread induced double positive stain index reduction has been mapped by generating RIM. This tool allows the appropriate assignment of fluorochromes to specificities to avoid dim signal loss of resolution due to bright signal spread.

Based on these principles:

(i) Dim markers have been assigned to fluorochromes that are spread free (or minimally impacted by spread) from co-expressed brilliant markers.
(ii) Bright markers have been positioned (when possible) to minimize the generation of spread into other colors.
(iii) Dim signals have been assigned to bright fluorochromes.
(iv) Bright signals have been preferentially assigned to dim fluorochromes.

## Titrations of mAbs and tetramers

Each fluorochrome-conjugated mAb, MAIT, and iNKT cell–specific tetramers listed in T-cell panel (Table S2) were titrated before performing multicolor stainings as follows.

As indicated in Table S3, mAbs were titrated with thawed ex vivo or in vitro PHA (1 $\mu$g/ml) activated PBMCs for 72 h. After thawing or activation, PBMCs were plated at 0.5 × 10$^6$ cells/well in a 96 U-bottomed well plate, pelleted and blocked in PBS + 10% NHS 10 $\mu$l/well. After 15 min at r.t., without washing, cells were incubated with 1:2 serial dilutions of mAb (from 0.1 to 6.4 $\mu$l) or tetramers (from 0.5 to 3 $\mu$l) in BD Horizon Brilliant Stain Buffer 50 $\mu$l/well. Some mAbs were titrated upon gating of PBMCs sub-populations identified by specific antibodies added to the titration mix as indicated in Table S3. mAbs were incubated for 20 min at r.t., whereas tetramers were incubated for 30 min at +4°C in the dark. In both cases, cells were washed three times at 510$g$ for 5 min in 200 $\mu$l FACS wash (PBS w/o Ca$^{2+}$ and Mg$^{2+}$ + 2% FCS, 0.2% NaN$_3$), then resuspended in FACS wash (150 $\mu$l/well). For vitality dyes titration: Thawed PBMCs were incubated at 65°C for 10 min in PBS to obtain dead cells. Live cells were added to dead cells at 1:1 ratio and 1 × 10$^6$ total (live and dead) cells were dispensed in 96 U-bottomed well plates, pelleted

and incubated with 1:2 serial dilutions of each of the vitality dye in 100 μl PBS. After 15 min in the dark at r.t, cells were washed once at 510$g$ for 5 min in 200 μl PBS and resuspended in FACS wash (150 μl/well). Cells stained with antibodies, tetramers, and vitality dyes were immediately analyzed by FACSymphony A5 flow cytometer (BD Biosciences). Cells acquired with serial dilution of reagents were concatenated by FlowJo software and the dilution leading to the best Stain Index was selected for multicolor staining.

## Procedure for fluorochrome spillover compensation

To calculate the compensation matrix and to correct the reciprocal spillover among fluorochromes with overlapping emission spectra, anti-mouse (BD CompBead Cat. no. 552843) or anti-rat (BD CompBead Cat. no. 552845) Ig-Kappa Comp Beads were stained accordingly to the manufacturer's conditions with the same mouse Igk or rat Igk anti-human antibodies used in the panels and immediately acquired using BD FACSDiva software (BD Biosciences version 8.0.2). The compensation matrix was automatically calculated by BD FACSDiva software with some minimal manual correction by FlowJo version 7.4.2. Slightly different compensation matrices have been applied to PBMCs and intratumoral T cells from the same patients stained with the same antibody panels. This can be due to different cell autofluorescence and fluorochrome interaction with specific sample components. FACS Symphony A5 flow cytometer performance was validated on a daily basis and the acquisition conditions were standardized by rainbow calibration particles (Spherotech Cat. no. RCP-30-5A) and CS&T beads (BD Biosciences Cat. no. 655051) following the manufacturer's instructions (Wang & Hoffman, 2017; Cossarizza et al, 2019).

## Peripheral blood and hepatic tissue metastases collection and processing

Peripheral blood samples, collected immediately before the surgical resection and hepatic tissue sections were obtained from Hepatobiliary Surgery at Ospedale San Raffaele with approval of the local Ethics Committee and written informed consent of donors in accordance with the Declaration of Helsinki. Licensed pathologists confirmed the histologic diagnosis of Colon Carcinoma derived Liver Metastases (CRC-LM) and collected sections from tumor tissue and matched peritumoral tissue at the front edge of the malignant lesion or normal liver tissue distal from the malignant lesion.

### Blood processing
PBMCs were isolated from healthy donors (buffy coats) by Ficoll (Cytiva Cat. no. 17-1440-03) density separation and activated in vitro with 1 μg/ml PHA (Remel–Invitrogen Cat. no. R30852801) in 24-well plates (Nunc 142485) (2 × 10⁶ cells in 2 ml complete RPMI 1640 Gibco Cat. no. 61870036 + 10% FCS Euroclone cat ECS1102L/well) for 72 h at 37°C + 5% $CO_2$.

Pre-surgical whole peripheral blood was processed by red blood cell lysis by incubation in Red Blood Cell Lysis Solution (Miltenyi Biotec Cat. no. 130-094-183), according to the manufacturer's instructions.

### CRC-LM tissue processing
(1) CRC-LM samples (normal liver, peritumoral, and intratumoral tissue sections) were collected immediately after surgical resection in 15 ml Falcon tubes containing 3 ml of MACS Tissue Storage Solution (Miltenyi Biotec Cat. no. 130-100-008) to preserve cell viability and stored at 4°C for no more than 20 h.

(2) The tissue sections were dried by tapping on filter paper, weighted, and divided in portions (up to 300 mg each) which were separately fragmented with scalpels into about 2-mm³ pieces in presence of the tissue storage solution in a tissue culture dish.

(3) Each ≤ 300-mg tissue section was separately transferred together with the tissue storage solution into one gentleMACS C Tube (Miltenyi Biotec Cat. no. 130-093-237) containing 4.7 ml complete RPMI 1640 and kept on ice.

(4) Once all tissue samples were minced, Enzyme H + Enzyme A (Tumor Dissociation Kit Human; Miltenyi Biotec Cat. no. 130-095-929 as described by manufacturer) + Collagenase CLSPA (22 U/ml final concentration) (Worthington Cat. no. LS005273), or for comparison Collagenase IV (Sigma-Aldrich Cat. no. C5138) (10 U/ml final concentration) or Dissociation kit Human Miltenyi Enzyme R (Miltenyi Biotec Cat. no. 130-095-929 as described by manufacturer) were added.

(5) GentleMACS C Tubes and heaters were installed in the gentleMACS Octo Dissociator (Miltenyi Biotec Cat. no. 130-096-427) and the 37°C_Multi_A_01 program was selected to get the maximal tumor-infiltrating viable lymphoid and myeloid cell number.

(6) The processed tissue samples were filtered through a 70-μm cell strainer (Corning Cat. no. 352350) in a 50 ml Falcon tube and RPMI 1640 medium was added up to 50 ml.

(7) After centrifugation (300$g$, 7 min), the cell suspension was incubated in Red Blood Cell Lysis Solution (Miltenyi Biotec Cat. no. 130-094-183), according to the manufacturer's instructions; finally, the cells were washed at 300$g$ for 7 min, resuspended in 2 ml PBS (Euroclone Cat. no. ECB4004L), and counted on a microscope.

(8) Cells were stained as described below with the following antibodies: anti-CD3 APC-H7 (560275; BD), anti-CXCR3 PE-CF594 (353736; BioLegend), anti-CXCR5 BB515 (564625; BD), anti-CCR6 BB790 (BD custom product), anti-CCR7 BV711 (566602; BD), and anti-CD25 BUV563 (612919; BD).

## Cell staining procedure for T-cell surface antigens

PHA-activated (1 × 10⁶ cells) PBMCs or leukocytes infiltrating CRC-LM (0.5–1 × 10⁶ tumor-infiltrating myeloid + lymphoid cells) were stained with titrated (Table S2) Fixable Viability Stain 620 (FVS620) (BD Biosciences Cat. no. 564996) in 100 μl PBS.

(1) After 15 min at r.t. in the dark, the cells were washed at 510$g$ for 5 min with PBS, transferred in a 96 U-bottomed well plate (Greiner Bio-One Cat. no. 650101) and incubated with Human Fc Block (BD Biosciences Cat. no. 564220) 5 μl/sample in 10 μl FACS wash (PBS + 2% FCS, 0.2% NaN₃ Sigma-Aldrich Cat. no. S2002) for 15 min at r.t. in the dark.

(2) Without further washing, CD1d and MR1 cell titrated tetramers (provided by NIH core tetramer facility hMR1 5-OP-RU 1.5 μg/ml,

hCD1d PBS-57 1.1 $\mu$g/ml) were added in 50 $\mu$l BD Horizon Brilliant Stain Buffer (BD Biosciences cat 566349)/sample and incubated for 30 min on ice in the dark.

(3) After two washings at 510$g$ for 5 min in 200 $\mu$l FACS wash, the cells were incubated back with Human Fc Block 2.5 $\mu$l/sample in 10 $\mu$l FACS wash for 15 min at r.t., in the dark.

(4) mAbs were pre-mixed (as indicated in Table S1) in BD Horizon Brilliant Stain Buffer 50 $\mu$l/sample. To discard possible mAb aggregates, the mAb mix was centrifuged at 14,000$g$ for 1 min at r.t.

After blocking described in 3. without washing, cells were incubated with the mAb mix for 20 min at r.t., washed three times as above described, resuspended in FACS wash (200 $\mu$l/well) and immediately acquired with FACSymphony A5 flow cytometer (BD Biosciences).

## Data analysis

Manual gating for flow cytometry data analysis was performed by FlowJo Software version 10.7.2 (BD Biosciences) on live cells after the exclusion of cell doublets (Fig S5). HD analysis was performed by cytoChain web application (Manfredi et al, 2021), briefly:

(1) To exclude signal fluctuations, data were cleaned by FlowAI (Monaco et al, 2016) present among FlowJo plugins, but also among cytoChain pre-analytical steps. The cleaned CRC-LM T-cell population was gated on the basis of CD45$^+$CD3$^+$CD19$^-$ expression from total lymphocytes excluding dead and aggregated cells. All the compensated parameters expressed in bi-exponential scale were exported by using "Export" FlowJo option as .FCS file, including "Time" among the parameters.

(2) The exported .FCS file was uploaded into cytoChain and SPADE correction (Qiu et al, 2011) was applied as a further pre-analytical cleaning step to avoid multidimensional data scattering.

(3) Data were transformed by Archsin scale.

(4) If multiple .FCS files have to be analyzed, they can be down-sampled by cytoChain to both reduce and/or balance the number of events to be analyzed among different samples.

(5) We applied the dimensionality reduction process based on t-SNE algorithm (Mahfouz et al, 2015) excluding parameters associated to dead cell fluorescent dye, CD45, CD3 and CD19 markers. If multiple .FCS files are uploaded they must be merged (concatenated) before the dimensionality reduction process.

(6) Clustering on the reported CRC-LM data set was run both with by FlowSOM (Van Gassen et al, 2015) and FastPG (Bodenheimer et al, 2020 Preprint) algorithms. In our case, we estimated that FastPG outperformed FlowSOM.

(7) Final graphical output was released by cytoChain to visualize, as described in the main text, the analysis results.

(8) We re-exported the FCS. file analyzed by cytoChain (the same can be done for concatenated .FCS files) and uploaded back into the FlowJo software for further visualization options and/or phenotype cluster de-convolution with typical histograms describing the fluorescence intensity profile for each antigen of interest. The re-exported files contain new dimensions describing the t-SNE bi-dimensional distribution and the cluster identity. If a concatenate of different samples is re-exported, single .FCS files can be retrieved and associated to t-SNE maps or clusters.

(9) The cell frequency contained in each cluster can be compared among different cell subpopulations or samples for statistical evaluation.

## Supplementary Information

## Acknowledgements

This study was funded by Italian Ministry of University and Research (PRIN 2017WC8499), Italian Healthy Ministry project on CAR T RCR-2019-23669115 to C Bonini, G Casorati, P Dellabona, and by Italian Association for Cancer Research (AIRC; IG 18458) to C Bonini, and by AIRC5x1000 (22737) to C Bonini, P Dellabona and MP Protti.

### Author Contributions

C Faccani: data curation, formal analysis, investigation, and methodology.
G Rotta: investigation, methodology, formal analysis, and writing—original draft.
F Clemente: formal analysis, validation, and investigation.
M Fedeli: methodology.
D Abbati: software and formal analysis.
F Manfredi: software.
A Potenza: methodology.
A Anselmo: methodology.
F Pedica: data curation, investigation, and histologic diagnosis.
G Fiorentini: data curation and surgery.
C Villa: resources and methodology.
MP Protti: methodology and funding acquisition.
C Doglioni: data curation, investigation, and histologic diagnosis.
L Aldrighetti: data curation, investigation, and surgery.
C Bonini: conceptualization, supervision, and funding acquisition.
G Casorati: conceptualization, supervision, funding acquisition, and writing—review and editing.
P Dellabona: conceptualization, supervision, funding acquisition, and writing—review and editing.
C de Lalla: conceptualization, formal analysis, supervision, investigation, methodology, and writing—original draft, review, and editing.

### Conflict of Interest Statement

The authors declare that they have no conflict of interest.

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
