## [Reviewer comments · Life Science Alliance]

Life Science Alliance

Workflow for high dimensional flow cytometry analysis of T cells from tumor metastases

Claudia de Lalla, Cristina Faccani, Gianluca Rotta, Francesca Clemente, Maya Fedeli, Danilo Abbati, Francesco Manfredi, Alessia Potenza, Achille Anselmo, Federica Pedica, Guido Fiorentini, Chiara Villa, Maria Protti, Claudio Doglioni, Luca Aldrighetti, Chiara Bonini, Giulia Casorati, and Paolo Dellabona

DOI: <https://doi.org/10.26508/lsa.202101316>

Corresponding author(s): *Claudia de Lalla, Ospedale San Raffaele Scientific Institute*

Review Timeline:

Submission Date:	2021-11-24
Editorial Decision:	2022-01-24
Revision Received:	2022-04-27
Editorial Decision:	2022-05-19
Revision Received:	2022-05-24
Accepted:	2022-05-25

Scientific Editor: Novella Guidi

Transaction Report:

January 24, 2022

Re: Life Science Alliance manuscript #LSA-2021-01316

Dr. Claudia de Lalla
Ospedale San Raffaele Scientific Institute
Department of Immunology, Transplantation and Infectious Disease
Via Olgettina 58
Milano 20132
Italy

Dear Dr. de Lalla,

Thank you for submitting your manuscript entitled "Workflow for high dimensional flow cytometry of T cells in Colon Carcinoma-derived Liver Metastases" to Life Science Alliance. The manuscript was assessed by expert reviewers, whose comments are appended to this letter. We, thus, encourage you to submit a revised version of the manuscript back to LSA that responds to all of the reviewers' points.

Thank you for this interesting contribution to Life Science Alliance. We are looking forward to receiving your revised manuscript.

Sincerely,

B. MANUSCRIPT ORGANIZATION AND FORMATTING:

Reviewer #1 (Comments to the Authors (Required)):

In this manuscript, Faccani et al describe methods for isolation of tumor-infiltrating T cells and optimal detection of a surface marker panel using high resolution flow cytometry. The authors present a new way to optimize instrument settings, evaluate different enzymatic digestion protocols with regard to protection of surface molecules, suggest a T cell panel for evaluation of tumor-infiltrating T cells, and apply a newly developed analysis tool to the data generated, focusing on unconventional T cell populations. The article is generally easy to follow, and the different steps of the methodology well described. In certain parts, however, not least the discussion, the English language could be improved. I also have a few specific point where the manuscript could be improved.

It is generally unclear how many times each experiment was repeated, as only representative images are shown. Please indicate this in the text or the figure legend. Some experiments were apparently only performed in a single patient. I would recommend repeating these experiments with more patients, in order to confirm the findings.

The CD19 staining is not optimal in liver tissue (Fig. 5) compared to PBMC (Fig. 2). Maybe the isolation protocol removes epitopes of CD19. This should be discussed in the manuscript.

The definition of MAIT cells based only on tetramer staining is not optimal, one would like to see CD161 staining as well to confirm their identity. This is illustrated in Fig. 7A, where a relatively high proportion of putative MAIT cells express CD4. This is not in agreement with previous studies (doi:10.1111/imcb.12021, doi:10.1073/pnas.1812273115), and should be discussed. This problem is also illustrated in Fig. 7C-D, where all putative MAIT cells in cluster #6 are CD4+ and also have a lower tetramer expression than cells in other clusters. I find it questionable if cluster #6 actually contains very many MAIT cells.

Gammadelta T cells contain several different types with more or less innate features, which can be partly delineated based on Vgamma and Vdelta chain usage. I would recommend discuss this issue in some more detail in the manuscript, with an explanation as to why this definition was not made, as the authors specifically aimed at studying unconventional T cell subsets.

Reviewer #2 (Comments to the Authors (Required)):

In this paper Guidi and colleagues describe a carefully designed and executed workflow and analysis pipeline for a high-dimensional flowcytometry panel for T cell analysis in primary CRC tumors and synchronous liver mets. In general the different steps are carefully described in detail and results clearly presented. As such this methodological paper should be of value to the field. I only have a few minor remarks:

- 1) it is not always clear on how many samples the shown data are based. Rather than showing just representative examples, it would have been useful and instructive to also show the level of variability between samples and if the pipeline required tweaking to accommodate these. The analyzed 2-3 samples are rather minimal in this respect and should be extended to approximately n=6 samples.
- 2) The paper places a lot of emphasis on the minor invariant T cell subsets as a demonstration of the resolution of the employed analysis pipeline; it would have been good to also show the analysis workflow for the more conventional T cell subsets and their analysis in relation to multiple costimulatory, activation and immune checkpoint markers. It would have revealed how to handle analysis of such a complex high-dimensional panel in a structured way to arrive at an organized and useful data set; this is proving a huge hurdle in current analyses and would have been very instructive and helpful, raising the impact of the study.

Reviewer #3 (Comments to the Authors (Required)):

Changes in liver lymphocyte phenotype, number and activation status are indeed emerging to be of clinical significance in colorectal liver metastases. This paper therefore addresses an important topic. However, in its present form it is not clear what

this paper is trying to communicate. Is it a 'technical note' or a description of T cell phenotypes in liver metastases? The title says 'Workflow' but the abstract fails to summarise the main content of the paper. The authors are recommended to write the former first using data that is now 'hidden' in 'supplementary material' and then use their technique/experimental approach to gather useful and novel data from patient material, that clearly moves the field on from the significant contributions already made by Jerome Galon and others. However, if presenting this in the form of a technical note or report, the precise novel contributions need to be described more clearly, with data demonstrating their improvement on previously published techniques.

For example, the paper describes optimisation of a flow cytometry panel. Indeed, the 26 colour panel is impressive but the optimisation is not anything new unless the specific limitations to performing high-dimensional flow cytometry of cell populations isolated from liver tissue are described and overcome.

The 'appropriate unsupervised computational data analysis' is poorly elaborated. Even if the group has already published another paper about the software, technical detail needs to be provided and described here.

The paper also speaks about optimisation of tissue digestion to obtain viable leukocytes from liver tissue but it is not clear what additional steps have been added to the published techniques or why.

Here, rare unconventional T cell populations are described in the liver eg gdT cells, iNKTs and MAITs. They have been identified before but now include interesting sub-populations - and certainly changed phenotypes depending on the TME. Perhaps focusing on how these can be better described and quantified using this technical approach might make a more publishable paper.

Some of the material in this paper would be more relevant and useful to the field if data from a patient cohort were collected, analysed and added.

It is not acceptable in its present form.

Minor Issues

What is a 'Summary Blurb'? Is it supposed to be a lay summary? This single long sentence with several complex concepts and terms is very confusing for a scientist - it would be impossible for a lay person.

The large number of grammatical, linguistic and stylistic errors is disconcerting eg

- a. indiscriminate use of capital letters throughout eg A Workflow for high dimensional flow cytometry of T cells in Colon Carcinoma-derived Liver Metastases
- b. lack of line numbering throughout
- c. inaccurate Running title 'Dissection of T cells in liver metastasis' - this paper does not describe dissection of T cells
- d. Pg 2 'The in-depth dissection of T cells within the tumor microenvironment by High Dimensional (HD) Flow Cytometry is crucial to unveil the mechanism...'
- e. Pg 4 'Furthermore, the metastatic spread of primary cancers to distal organs, which are the main cause of death for cancer,

Dear Editors

Please find enclosed the point by point reply for the revised version of the manuscript entitled **"Workflow for high dimensional flow cytometry analysis of T cells from tumor metastases"** #LSA-2021-01316 by Cristina Faccani et al.
Thank you for your consideration.

On behalf of all authors
Sincerely
Claudia de Lalla

Point by point reply

We highlighted in yellow the major revisions included in the manuscript

Reviewer #1 (Comments to the Authors (Required)):

1. In this manuscript, Faccani et al describe methods for isolation of tumor-infiltrating T cells and optimal detection of a surface marker panel using high resolution flow cytometry. The authors present a new way to optimize instrument settings, evaluate different enzymatic digestion protocols with regard to protection of surface molecules, suggest a T cell panel for evaluation of tumor-infiltrating T cells, and apply a newly developed analysis tool to the data generated, focusing on unconventional T cell populations. The article is generally easy to follow, and the different steps of the methodology well described. In certain parts, however, not least the discussion, the English language could be improved. I also have a few specific point where the manuscript could be improved.

Reply. We thank the reviewer for the positive comments. The paper was completely revised to improve the English language.

2. It is generally unclear how many times each experiment was repeated, as only representative images are shown. Please indicate this in the text or the figure legend. Some experiments were apparently only performed in a single patient. I would recommend repeating these experiments with more patients, in order to confirm the findings.

Reply. We have included additional samples to support the findings, shown in the figures enclosed in Supplementary Figures: 1. One sample for the validation of Voltration procedure (Fig. S6); 2. One

sample for the analysis of antigen expression upon enzymatic digestion (Fig. S7); 3. Three samples for CRC-LM (Fig. S9-S11) and two samples for primary CRC (Fig. S12, S13).

3. The CD19 staining is not optimal in liver tissue (Fig. 5) compared to PBMC (Fig. 2). Maybe the isolation protocol removes epitopes of CD19. This should be discussed in the manuscript.

Reply. We found that the level of CD19 expression is independent of the enzymatic treatment during the tissue processing (Fig. 1 enclosed for the reviewer's perusal). It is likely that differences in CD19 expression level might be due to intrinsic characteristics of the samples analyzed and patient to patient variability.

4. The definition of MAIT cells based only on tetramer staining is not optimal, one would like to see CD161 staining as well to confirm their identity. This is illustrated in Fig. 7A, where a relatively high proportion of putative MAIT cells express CD4. This is not in agreement with previous studies (doi:10.1111/imcb.12021, doi:10.1073/pnas.1812273115), and should be discussed. This problem is also illustrated in Fig. 7C-D, where all putative MAIT cells in cluster #6 are CD4+ and also have a lower tetramer expression than cells in other clusters. I find it questionable if cluster #6 actually contains very many MAIT cells.

We respectfully disagree with the reviewer's comment on our MR1 tetramer staining and the detection of "bona fide" CD4+ MAIT cells. The MR1-5-OP-RU tetramers described in the manuscript were reported to be more adequate than staining for TRAV1-2 CD161 surrogate markers to identify MAIT cells (Nicholas A. Gherardin et al. Human blood MAIT cell subsets defined using MR1 tetramers. Immunol Cell Biol. 2018). In the same paper MR1tet+ CD4+ T cells are described as bona fide MAIT cells with distinct effector functions and a lower expression of CD161 than the CD4-CD8- and CD8+ cell subsets. Furthermore, a CD4+ MAIT cell subset expressing high levels of CD39 was found in T cells infiltrating primary CRC, supporting the possibility that this MAIT subpopulation could infiltrate also a metastatic lesion. (Li S. et al. Human Tumor-Infiltrating MAIT Cells Display Hallmarks of Bacterial Antigen Recognition in Colorectal Cancer. Cell Reports Medicine. 2020). In support to our conclusions, we show for the reviewer (Fig. 2 enclosed for the reviewer's perusal) that, according to the above studies, CD161 is differently distributed among CRC-LM infiltrating MAIT subsets, with the highest frequency of CD161+ among CD8+ and CD4-CD8- MAIT cells and the lowest among CD4+ MAIT cells. Moreover, it has been reported that caution must be applied in interpreting data relative to MAIT cells on the basis of CD161 expression, as this marker can be down-regulated in disease settings (Leeansyah E. et al. Activation, exhaustion, and persistent decline of the antimicrobial MR1-restricted MAIT-cell population in chronic HIV-1infection. Blood. 2013). We included these observations in the manuscript (Discussion).

5. Gammadelta T cells contain several different types with more or less innate features, which can be partly delineated based on Vgamma and Vdelta chain usage. I would recommend discuss this issue in some more detail in the manuscript, with an explanation as to why this definition was not made, as the authors specifically aimed at studying unconventional T cell subsets.

Reply. We thank the reviewer for remarking an issue that we have now clarified in the manuscript (Discussion). Main aim of this study was to set up a workflow able to assess also $\gamma\delta$ T cells from liver

metastases. Now that we know that we can clearly detect them, we will discuss why we did not dissect their $V\gamma$ and $V\delta$ usage here, and that we are going to do it in an ongoing study with a much larger CRC-LM patient cohort. We included these observations in the manuscript (Discussion).

Reviewer #2 (Comments to the Authors (Required)):

In this paper Guidi and colleagues describe a carefully designed and executed workflow and analysis pipeline for a high-dimensional flow cytometry panel for T cell analysis in primary CRC tumors and synchronous liver mets. In general the different steps are carefully described in detail and results clearly presented. As such this methodological paper should be of value to the field. I only have a few minor remarks:

Reply. We thank the reviewer for the favorable comments.

1) it is not always clear on how many samples the shown data are based. Rather than showing just representative examples, it would have been useful and instructive to also show the level of variability between samples and if the pipeline required tweaking to accommodate these. The analyzed 2-3 samples are rather minimal in this respect and should be extended to approximately $n=6$ samples.

Reply. We have included additional samples to support the findings, shown in the figures enclosed in Supplementary Figures: 1. One sample for the validation of Voltration procedure (Fig. S6); 2. One sample for the analysis of antigen expression upon enzymatic digestion (Fig. S7); 3. Three samples for CRC-LM (Fig. S9-S11) and two samples for primary CRC (Fig. S12, S13).

2) The paper places a lot of emphasis on the minor invariant T cell subsets as a demonstration of the resolution of the employed analysis pipeline; it would have been good to also show the analysis workflow for the more conventional T cell subsets and their analysis in relation to multiple costimulatory, activation and immune checkpoint markers. It would have revealed how to handle analysis of such a complex high-dimensional panel in a structured way to arrive at an organized and useful data set; this is proving a huge hurdle in current analyses and would have been very instructive and helpful, raising the impact of the study.

Reply. We agree with the reviewer's remark and revised the manuscript accordingly. We introduced Figure 8 in the main text of the manuscript showing the phenotype of the clusters identified by the FastPhenograph algorithm among the whole T cell population infiltrating a CRC-LM. This gave us the opportunity to show a comprehensive approach of high dimensional analysis including 1. Analysis of tSNE maps 2. Application of an algorithm for cell clustering, 3. Analysis and visualization at a glance of the cell phenotype for each cluster contained in the whole T cell population by a heat-map, 4. Quantification of the cell frequency within each cell cluster. 5. Integration of classical manual gating and unsupervised computational analysis of unconventional T cells 6. Deconvolution of the phenotype of rare cell clusters (Fig.9 main text).

Reviewer #3 (Comments to the Authors (Required)):

1. Changes in liver lymphocyte phenotype, number and activation status are indeed emerging to be of clinical significance in colorectal liver metastases. This paper therefore addresses an important topic. However, in its present form it is not clear what this paper is trying to communicate. Is it a 'technical note' or a description of T cell phenotypes in liver metastases? The title says 'Workflow' but the abstract fails to summarise the main content of the paper. The authors are recommended to write the former first using data that is now 'hidden' in 'supplementary material' and then use their technique/experimental approach to gather useful and novel data from patient material, that clearly moves the field on from the significant contributions already made by Jerome Galon and others. However, if presenting this in the form of a technical note or report, the precise novel contributions need to be described more clearly, with data demonstrating their improvement on previously published techniques.

Reply. We thank the reviewer for the comments and rephrased the entire manuscript in order to convey the clear message that this is a methodological note. We believe we have clarified it; the relevant steps of the workflow are described in the main text of the manuscript and supplementary materials contain technical details to support our results.

2. For example, the paper describes optimisation of a flow cytometry panel. Indeed, the 26 colour panel is impressive but the optimisation is not anything new unless the specific limitations to performing high-dimensional flow cytometry of cell populations isolated from liver tissue are described and overcome.

Reply. We thank the reviewer for the positive comment about the quality of our 26-color based flow cytometry analysis. This is the result of a novel instrumental setting procedure based on the use of PBMCs with the same autofluorescence of the immune cells isolated from tissues and analysed by the multiparametric panel allowing a better signal resolution. We also show the problems raised by isolating intact T lymphocytes from metastatic liver tissue and the methods we implemented to overcome them. Furthermore, to our knowledge, our study is the first to describe a panel targeting activation and exhaustion markers not only for CRC-LM infiltrating conventional T cells but also unconventional T cells identified by antigen-bound tetramers.

3. The 'appropriate unsupervised computational data analysis' is poorly elaborated. Even if the group has already published another paper about the software, technical detail needs to be provided and described here.

Reply. We have thoroughly revised the manuscript to improve the description of the 'appropriate unsupervised computational data analysis'. The Results section describes in greater details the cytoChain pipeline, and the Material and Methods section details the pre-analytical and analytical steps of the pipeline that were adopted. We introduced Figure 8 in the main text of the manuscript showing the phenotype of the clusters identified by the FastPhenograph algorithm among the whole T cell population infiltrating a CRC-LM. This gave us the opportunity to show a comprehensive approach of high dimensional analysis including 1. Analysis of tSNE maps 2. Application of an algorithm for

cell clustering, 3. Analysis and visualization at a glance of the cell phenotype for each cluster contained in the whole T cell population by a heat-map, 4. Quantification of the cell frequency within each cell cluster. 5. Integration of classical manual gating and unsupervised computational analysis of unconventional T cells 6. Deconvolution of the phenotype of rare cell clusters (Fig.9 main text).

4. The paper also speaks about optimisation of tissue digestion to obtain viable leukocytes from liver tissue but it is not clear what additional steps have been added to the published techniques or why.

Reply. We have carefully tested and fine-tuned several commercial and non-commercial tissue digestion methods, none of which was specifically indicated to retrieve intact T cells from cancer metastases in the liver, until we have defined an optimal and novel recipe to achieve so. The whole procedure followed two steps:

- 1. The commercially available enzyme cocktail R from Miltenyi, which is widely utilized to digest cancer tissues from different origins, could not be used for liver metastases from CRC, as this reagent destroyed antigens on the T cell surface even at very low concentrations.*
- 2. Hence, we tested other enzymatic cocktails for tissue extraction and identified optimal performance for CLSPA, previously reported to better preserve the surface antigens expressed by immune cells isolated from other tissues.*

5. Here, rare unconventional T cell populations are described in the liver eg gdT cells, iNKTs and MAITs. They have been identified before but now include interesting sub-populations - and certainly changed phenotypes depending on the TME. Perhaps focusing on how these can be better described and quantified using this technical approach might make a more publishable paper.

Reply. We describe the unconventional T cells in Fig.9 and discuss the result obtained with our new workflow that allows an accurate dissection of these populations.

6. Some of the material in this paper would be more relevant and useful to the field if data from a patient cohort were collected, analysed and added.

Reply. This paper was not aimed at providing biological results on LM-CRC immune-infiltrates, but rather at defining an accurate method to analyse T cells with a reproducible experimental approach and robust analysis. An ongoing study on a large cohort of prospectively collected patients will eventually provide novel biological information to be associated with clinical characteristics of the patients.

Minor

Issues

1. What is a 'Summary Blurb'? Is it supposed to be a lay summary? This single long sentence with several complex concepts and terms is very confusing for a scientist - it would be impossible for a lay person.

Reply. The Summary Blurb is requested by the Journal, to answer to the reviewer's comment we attempted to simplify it as much as possible.

3. The large number of grammatical , linguistic and stylistic errors is disconcerting eg

Reply. The paper was thoroughly revised to amend all errors.

4. a. indiscriminate use of capital letters throughout eg A Workflow for high dimensional flow cytometry of T cells in Colon Carcinoma-derived Liver Metastases

b. lack of line numbering throughout

Reply. We amended the capital letters and added the line numbering.

c. inaccurate Running title 'Dissection of T cells in liver metastasis' - this paper does not describe dissection of T cells

Reply. We changed the running title with: " T cell phenotyping in liver metastases" (40 characters including spaces)

d. Pg 2 'The in-depth dissection of T cells within the tumor microenvironment by High Dimensional (HD) Flow Cytometry is crucial to unveil the mechanism...'

e. Pg 4 'Furthermore, the metastatic spread of primary cancers to distal organs, which are the main cause of death for cancer,

Reply. We corrected the sentences

Fig.1 CD19 molecule expression profile by lymphocyte from CRC-LM peritumor section in the presence or in the absence of collagenase enzymatic treatment. Peritumor surrounding a CRC-LM lesion and whole peripheral blood were processed from the same patient as described in MATERIAL & METHODS. CD19⁺ cells were gated and their frequency among peritumor total CD45⁺ cells obtained in the presence or in the absence of CLSPA treatment during the tissue mechanical dissociation was compared to the frequency of the same cell population circulating in the peripheral blood.

Fig.2 CD161 molecule expression profile by CRC-LM infiltrating MAIT cells analyzed by flow cytometry: A. CRC-LM infiltrating T cell gating strategy. **B.** T cells were stained in absence of anti-CD161 MoAb as negative control (left) **C.** Expression of CD161 molecule by CD4⁺, CD8⁺ and CD4⁻CD8⁻ (DN) subsets among MR1Tet⁺ MAIT cells .

May 19, 2022

RE: Life Science Alliance Manuscript #LSA-2021-01316R

Dr. Claudia de Lalla
Ospedale San Raffaele Scientific Institute
Department of Immunology, Transplantation and Infectious Disease
Via Olgettina 58
Milano 20132
Italy

Dear Dr. de Lalla,

Thank you for submitting your revised manuscript entitled "Workflow for high dimensional flow cytometry analysis of T cells from tumor metastases". We would be happy to publish your paper in Life Science Alliance pending final revisions necessary to meet our formatting guidelines.

- please add the Twitter handle of your host institute/organization as well as your own or/and one of the authors in our system
- please add a callout for Figure S5A and Figure S8 to your main manuscript text
- please add the appropriate panels that are listed in the figure legend to Figure 3
- please include supp. Materials and methods into the main Material and Methods section
- please provide separate figure legends for supp. Figs. 9-13

A. FINAL FILES:

B. MANUSCRIPT ORGANIZATION AND FORMATTING:

Sincerely,

Reviewer #1 (Comments to the Authors (Required)):

The manuscript has been considerably improved, and I don't have any further comments.

Reviewer #3 (Comments to the Authors (Required)):

This paper is significantly improved. It is now ready for publication.

May 25, 2022

RE: Life Science Alliance Manuscript #LSA-2021-01316RR

Dr. Claudia de Lalla
Ospedale San Raffaele Scientific Institute
Department of Immunology, Transplantation and Infectious Disease
Via Olgettina 58
Milano 20132
Italy

Dear Dr. de Lalla,

Thank you for submitting your Methods entitled "Workflow for high dimensional flow cytometry analysis of T cells from tumor metastases". It is a pleasure to let you know that your manuscript is now accepted for publication in Life Science Alliance. Congratulations on this interesting work.

DISTRIBUTION OF MATERIALS:

Again, congratulations on a very nice paper. I hope you found the review process to be constructive and are pleased with how the manuscript was handled editorially. We look forward to future exciting submissions from your lab.

Sincerely,
